# Measurement report: Method for evaluating $CO_2$ emissions from a cement plant using atmospheric $\delta(O_2 / N_2)$ and $CO_2$ measurements and its implication for future detection of $CO_2$ capture signals

**Shigeyuki Ishidoya[1], Kazuhiro Tsuboi[2], Hiroaki Kondo[1], Kentaro Ishijima[2], Nobuyuki Aoki[1], Hidekazu Matsueda[2], and Kazuyuki Saito[3]**

[1]Environmental Management Research Institute, National Institute of Advanced Industrial Science and Technology (AIST), Tsukuba 305-8569, Japan
[2]Department of Climate and Geochemistry Research, Meteorological Research Institute, Tsukuba 305-0052, Japan
[3]Atmosphere and Ocean Department, Atmospheric Environment and Ocean Division, Japan Meteorological Agency, Tokyo 105-8431, Japan

**Correspondence:** Shigeyuki Ishidoya (s-ishidoya@aist.go.jp)

**Abstract.** Continuous observations of atmospheric $\delta(O_2/N_2)$ and $CO_2$ amount fractions have been carried out at Ryori (RYO), Japan, since August 2017. In these observations, the $O_2 : CO_2$ exchange ratio (ER, $-\Delta y(O_2)\Delta y(CO_2)^{-1}$) has frequently been lower than expected from short-term variations in emissions from terrestrial biospheric activities and combustion of liquid, gas, and solid fuels. This finding suggests a substantial effect of $CO_2$ emissions from a cement plant located about 6 km northwest of RYO. To evaluate this effect quantitatively, we simulated $CO_2$ amount fractions in the area around RYO by using a fine-scale atmospheric transport model that incorporated $CO_2$ fluxes from terrestrial biospheric activities, fossil fuel combustion, and cement production. The simulated $CO_2$ amount fractions were converted to $O_2$ amount fractions by using the respective ER values of 1.1, 1.4, and 0 for the terrestrial biospheric activities, fossil fuel combustion, and cement production. Thus obtained $O_2$ and $CO_2$ amount fraction changes were used to derive a simulated ER for comparison with the observed ER. To extract the contribution of $CO_2$ emissions from the cement plant, we used $y(CO_2^*)$ as an indicator variable, where $y(CO_2^*)$ is a conservative variable for terrestrial biospheric activities and fossil fuel combustion obtained by simultaneous analysis of observed $\delta(O_2/N_2)$ and $CO_2$ amount fractions and simulated ERs. We confirmed that the observed and simulated ER values and also the $y(CO_2^*)$ values and simulated $CO_2$ amount fractions due only to cement production were generally consistent. These results suggest that combined measurements of $\delta(O_2/N_2)$ and $CO_2$ amount fractions will be useful for evaluating $CO_2$ capture from flue gas at carbon capture and storage (CCS) plants, which, similar to a cement plant, change $CO_2$ amount fractions without changing $O_2$ values, although CCS plants differ from cement plants in the direction of $CO_2$ exchange with the atmosphere.

# 1   Introduction

Simultaneous analysis of atmospheric $\delta(O_2/N_2)$ and $CO_2$ amount fractions has been used to estimate the global $CO_2$ budget since the early 1990s (e.g., Keeling and Shertz, 1992). Recently, these analyses have also been applied to separate the contributions of different sources to the local $CO_2$ budget in an urban area (e.g., Ishidoya et al., 2020; Sugawara et al., 2021; Pickers et al., 2022; Liu et al., 2023). This approach uses $-O_2 : CO_2$ exchange ratios (ER) or oxidative ratios (OR) $(-\Delta y(O_2)\Delta y(CO_2)^{-1})$ for terrestrial biospheric activities and fossil fuel combustion. Strictly speaking, there is a distinction between ERs and ORs; the ER refers to the exchange between the atmosphere and organisms or ecosystems while the OR indicates the stoichiometry of specific materials (Faassen et al., 2023). For terrestrial biospheric $O_2$ and $CO_2$ fluxes, ORs of 1.1 or 1.05 are generally used (Severinghaus, 1995; Resplandy et al., 2019), and for the fluxes due to fossil fuel combustion, ORs of 1.95 for gaseous fuels, 1.44 for oil and other liquid fuels, 1.17 for coal and other solid fuels, and 0 for cement production are typical (Keeling, 1988). Therefore, the atmospheric $O_2$ amount fraction varies in opposite phase with the $CO_2$ amount fraction, owing to terrestrial biospheric activities and fossil fuel combustion. The ORs are typically very stable, and the global average OR for fossil fuels is about 1.4 (e.g., Keeling and Manning, 2014).

In the cement production process, calcium carbonate is burned and calcium oxide and $CO_2$ are produced as follows:

$$CaCO_3 \rightarrow CaO + CO_2. \tag{R1}$$

Because this chemical reaction emits $CO_2$ to the atmosphere without $O_2$ consumption, its OR is 0. It should be noted that the cement kilns are usually fired with fossil fuels, so that the overall ER for cement production is not 0. $CO_2$ emissions from cement production account for about 2 % of global fossil fuel $CO_2$ emissions (Friedlingstein et al., 2022). However, because it is difficult to separate the cement production signal from $CO_2$ emissions due to fossil fuel combustion and terrestrial biospheric activities, no study has reported direct evidence of variations in the atmospheric $CO_2$ amount fraction due to cement production at the Global Atmosphere Watch (GAW) program of the World Meteorological Organization (WMO) stations. In this context, simultaneous observations of $\delta(O_2/N_2)$ and $CO_2$ amount fractions are expected to be useful for separating out the cement production signal owing to its characteristic OR value. Moreover, Keeling et al. (2011), who examined the possibility of verifying rates of carbon capture and storage (CCS) and direct air capture of $CO_2$ (DAC) by using changes in the atmospheric constituents, suggested that combined measurements of the $\delta(O_2/N_2)$ and $CO_2$ could powerfully constrain estimated rates.

To investigate $CO_2$ leak detection from a CCS site, van Leeuwen and Meijer (2015) observed $\delta(O_2/N_2)$ and $CO_2$ from a 6 m tall mast that was 5–15 m away from artificial $CO_2$ release points. They estimated that their measurement system could detect a $CO_2$ leak of $10^3$ t a$^{-1}$ at a location up to 500 m away from the leak point. Pak et al. (2016) monitored the air for $CO_2$ plumes at locations between 1 and 100 m from an artificial $CO_2$ release point, and collected air samples typically between 9 and 20 m from the point where the $CO_2$ amount fraction was 100–600 $\mu$mol mol$^{-1}$ above ambient. They then analyzed the air samples for $O_2$ and $CO_2$ amount fractions and found much lower ERs than those expected from fossil fuel combustion and terrestrial biospheric activities. These studies support the suggestion by Keeling et al. (2011) regarding the usefulness of $\delta(O_2/N_2)$ and $CO_2$ measurements. As the next step to verify the usefulness of combined measurements of $\delta(O_2/N_2)$ and $CO_2$, their applicability to the detection of not only $CO_2$ leaks but also $CO_2$ capture from flue gas should be examined. In this regard, CCS/DAC plants remove $CO_2$ from the atmosphere without causing any $O_2$ changes, just as cement plants do, differing only in the direction of $CO_2$ exchange between the plant and the atmosphere. Therefore, it should be possible to evaluate the ability of combined measurements to detect a $CO_2$ capture signal by showing that they can be used to detect a cement production signal.

In this paper, we present evidence of the successful detection of a cement production signal by combined measurements of $\delta(O_2/N_2)$ and $CO_2$ at a ground station (a designated WMO/GAW local site) located near a cement plant. We also examine the usefulness of the measurements for future detection of CCS/DAC signals by using a fine-scale 3-D atmospheric transport model to investigate the consistency between the observed signal and the simulated $CO_2$ emissions from the plant.

## 2   Methods

### 2.1   Observations of atmospheric $\delta(O_2 / N_2)$ and $CO_2$ amount fractions

Atmospheric $\delta(O_2/N_2)$ and $CO_2$ amount fractions have been observed continuously at the coastal station Ryori (RYO: 39°2' N, 141°49' E, 260 m a.s.l.; Fig. 1), Japan, since 2017, by using a paramagnetic $O_2$ analyzer (POM-6E, Japan Air Liquid) and a non-dispersive infrared $CO_2$ analyzer (NDIR; LI-7000, LI-COR), respectively. RYO is a designated WMO/GAW station, and the Japan Meteorological Agency (JMA) has also observed $CO_2$, $CH_4$, and CO amount fractions there since 1987, 1991, and 1991, respectively (e.g., Wada et al., 2011). The $CO_2$, $CH_4$, and CO amount fraction data observed by JMA are available online at the WMO World Data Centre for Greenhouse Gases (WMO/WDCGG; https://gaw.kishou.go.jp/, last access: 10 November 2023). A cement plant (Taiheiyo Cement Ofunato Plant) is 6 km away from RYO (Fig. 1). It should be noted that the $CO_2$ amount fraction data posted on WDCGG have already been classified into the data for background air and those affected by

local fossil fuel combustion including the cement production discussed in this study. The annual cement production at the plant is $1.966 \times 10^6 \, \text{t a}^{-1}$ (https://www.taiheiyo-cement.co.jp/english/index.html, 5 January 2024).

The $\delta(O_2/N_2)$ is reported in per meg, where 1 per meg is $0.001\,\text{‰}$:

$$\delta(O_2/N_2) = \frac{R_{\text{sample}}\left({}^{16}O^{16}O/{}^{14}N^{14}N\right)}{R_{\text{standard}}\left({}^{16}O^{16}O/{}^{14}N^{14}N\right)} - 1, \qquad (1)$$

where the subscripts "sample" and "standard" indicate the sample air and the standard gas, respectively. Because the $O_2$ amount fraction in dry air is $0.2093$–$0.2094 \, \text{mol mol}^{-1}$ (Tohjima et al., 2005; Aoki et al., 2019), the addition of $1 \, \mu\text{mol}$ of $O_2$ to $1 \, \text{mol}$ of dry air increases $\delta(O_2/N_2)$ by 4.8 per meg ($= 1/0.2094$). If $CO_2$ is converted one-for-one into $O_2$, it causes $\delta(O_2/N_2)$ to increase by 4.8 per meg, which is equivalent to an increase of $1 \, \mu\text{mol mol}^{-1}$ of $O_2$ for each $1 \, \mu\text{mol mol}^{-1}$ decrease in $CO_2$. Therefore, observed relative changes in $\delta(O_2/N_2)$ were converted to those in $O_2$ amount fraction by multiplying by $0.2094 \, \mu\text{mol mol}^{-1}$ (per meg)$^{-1}$.

In this study, $\delta(O_2/N_2)$ of each air sample was measured with a paramagnetic analyzer using high- and low-span standard air of which $\delta(O_2/N_2)$ had been measured against our primary standard air (Cylinder No. CRC00045; AIST-scale) using a mass spectrometer (Thermo Scientific Delta-V) (Ishidoya and Murayama, 2014). The scale based on the primary standard air is our original scale, called the "EMRI/AIST scale" in Aoki et al. (2021). Sample air was taken at the tower heights of $20 \, \text{m}$ using a diaphragm pump at a flow rate higher than $10 \, \text{L min}^{-1}$ to prevent thermally diffusive fractionation of air molecules at the air intake (Blaine et al., 2006). The tower is situated on the windward side of the prevailing wind direction, and the surface below the tower consists of short grass. Then, a large portion of the air is exhausted from the buffer, with the remaining air allowed to flow into the analyzers from the center of the buffer. It is then sent to an electric cooling unit with a water trap cooled to $-80\,°C$ at a flow rate of $100 \, \text{mL min}^{-1}$, with the pressure stabilized to $0.1 \, \text{Pa}$ and measured for $90 \, \text{min}$. After the measurements, high-span standard gas, prepared by adding appropriate amounts of pure $O_2$ or $N_2$ to industrially prepared $CO_2$ standard air, was introduced into the analyzers with the same flow rate and pressure as the sample air and measured for $5 \, \text{min}$, and low-span standard gas was then measured through the same procedure. The dilution effects on the $O_2$ mole fraction measured by the paramagnetic analyzer were corrected experimentally, not only for the changes in $CO_2$ of the sample air or standard gas measured by the NDIR, but also for the changes in Ar of the standard gas measured by the mass spectrometer as $\delta(Ar/N_2)$.

The analytical reproducibility of the $\delta(O_2/N_2)$ and $CO_2$ amount fraction measurements by the system was determined by repeated measurements of standard gas and found to be about 5 per meg and $0.06 \, \mu\text{mol mol}^{-1}$, respectively, for 2 min-average values. For more information, see Ishidoya et al. (2017). In this study, we use about 70 min-average mean values for analysis. It should be noted that gaps in the data seen at the end of August to beginning of September 2017 are due to maintenance and technical issues other than routine calibrations described earlier. The number of $\delta(O_2/N_2)$ (and $CO_2$ amount fraction) data points shown in Fig. 2 is 9220. Note that we used a mass spectrometer to measure both $\delta(O_2/N_2)$ and the $CO_2$ amount fraction of the working standard air, whereas we determined the $CO_2$ amount fraction on the TU-10 scale using a gravimetrically prepared air-based $CO_2$ standard gas system (Nakazawa et al., 1997). However, we found that the $CO_2$ amount fractions observed in this study were systematically higher by about $1 \, \mu\text{mol mol}^{-1}$ than those observed by JMA and reported on the WMO scale (X2007), which is larger than that expected from the scale difference of about $0.2 \, \mu\text{mol mol}^{-1}$ between the TU-10 and WMO scales (Tsuboi et al., 2016). This discrepancy might be related to the LI-7000 NDIR used in this study because no significant difference has been found between the TU-10 and WMO scales at Minamitorishima, where a different NDIR (LI-820, LI-COR) has been used for continuous measurements of $\delta(O_2/N_2)$ and $CO_2$ amount fractions (Ishidoya et al., 2017). However, we found no significant difference in span sensitivities between the $CO_2$ amount fractions observed in this study and those observed by JMA. Therefore, the systematic difference between the observed $CO_2$ amount fractions and those observed by JMA does not affect the ER values, discussed in Sect. 3, which were calculated from changes in $O_2$ and $CO_2$ amount fractions. The $\delta(O_2/N_2)$ and $CO_2$ amount fractions observed at RYO are available in the Supplement.

## 2.2 Simulation of atmospheric $CO_2$ and $O_2$ amount fractions using an atmospheric transport model

To calculate the local transport of $CO_2$ around RYO, we used the National Institute of Advanced Industrial Science and Technology (AIST) Mesoscale Model (AIST-MM) fine-scale regional atmospheric transport model (Kondo et al., 2001). AIST-MM is a one-way nested model with an outer domain that covers east Japan with an approximately $10 \, \text{km}$ grid interval and an inner domain that covers an area of $120 \, \text{km} \times 120 \, \text{km}$ near RYO with a grid interval of approximately $1 \, \text{km}$ (Fig. 1). The EAGrid2010-Japan emissions inventory (Fukui et al., 2014), an update of the EAGrid2000-Japan inventory (Kannari et al., 2007) to the year 2010, was used for fossil fuel combustion. In this study, fossil fuel combustion means anthropogenic $CO_2$ sources other than cement production. The spatial resolution of EAGrid2010-Japan is approximately $1 \, \text{km}$, and the temporal resolution is monthly average of $1 \, \text{h}$. No further inter-annual correction of emissions is employed, but EAGrid2010-Japan considers the difference in traffic volume between weekdays and holidays. To calculate the $CO_2$ budget for vegetation, the NCAR Land Surface Model (Bonan, 1996) was used as a

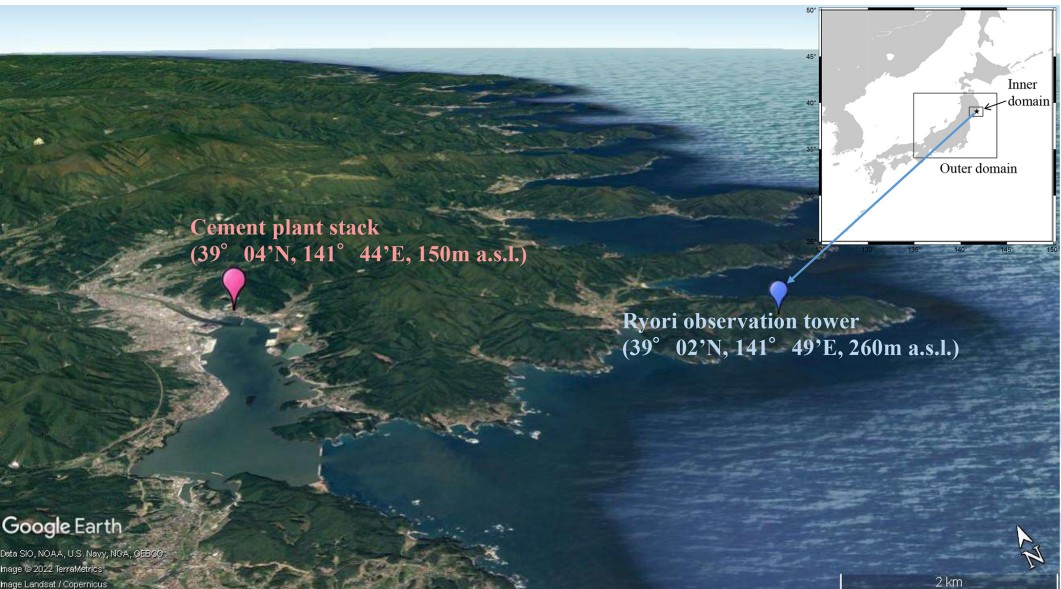

**Figure 1.** Location of the Ryori site (RYO) and the cement plant on an aerial photograph from © Google Earth. The cement plant is about 6 km northwest of RYO. Inner and outer domains of the fine-scale 3-D atmospheric transport model (AIST-MM) used in the present study are also shown.

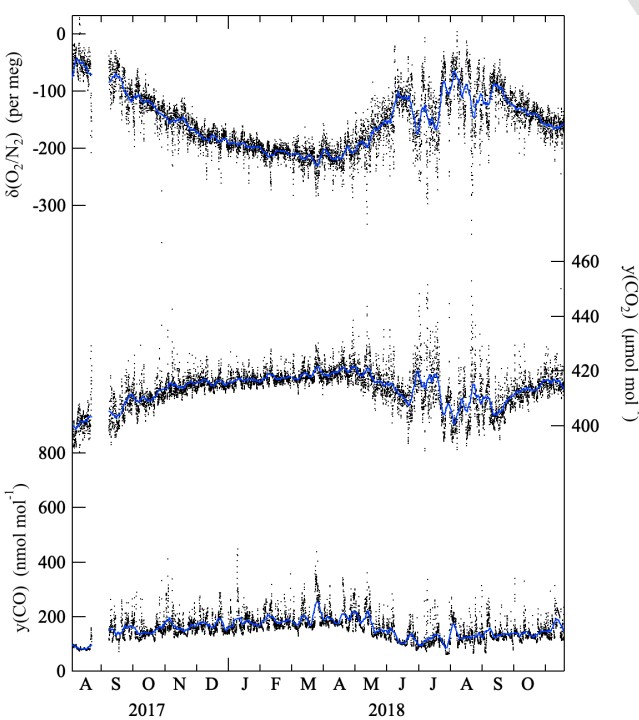

**Figure 2.** $\delta(O_2/N_2)$ and $CO_2$ and CO amount fractions (black dots) and their 1-week rolling average values (blue lines) observed at Ryori (RYO), Japan, from August 2017 to November 2018. $\delta(O_2/N_2)$ and $CO_2$ $y$ axes are scaled to be visually comparable.

sub-model, replacing the simple function of temperature and solar insolation used in the original AIST-MM for this calculation. The cement plant source was set at the location of the plant's stack, at the effective stack height of 275 m. The $CO_2$ emissions from the cement plant were estimated from the clinker production capacity of the Ofunato plant in 2018 (Japan Cement Association, 2020). The clinker is a solid material produced in cement manufacture as an intermediary product of Portland cement, mainly consisting of CaO, $SiO_2$, $Al_2O_3$, and $Fe_2O_3$. The annual emissions were calculated using the method of the Ministry of Environmental Protection (https://www.env.go.jp/earth/ondanka/ghg-mrv/methodology/material/methodology_2A1.pdf, last access: 5 January 2024, in Japanese) as

$$E = P \times F \times D, \tag{2}$$

where $E$ is the annual emissions of $CO_2$ from the cement plant ($t\,a^{-1}$), $P$ is the annual production capacity of the clinker at the cement plant ($t\,a^{-1}$), $F$ is the $CO_2$-to-clinker mass ratio of 0.516, and $D$ is the cement kiln dust of 1. For initial and boundary conditions, we used GPV/MSM (grid point value of meso-scale model) meteorological data of wind, temperature, and humidity from JMA (https://www.jma.go.jp/jma/en/Activities/nwp.html, last access: 5 January 2024). As a result, $CO_2$ amount fractions at RYO are calculated by summing up the contributions of the $CO_2$ amount fraction for fossil fuel combustion, terrestrial biospheric activities, and cement production. In this study, not only $CO_2$ amount fractions but also ERs are compared between the observed and simulated data. For this purpose, $O_2$ amount fractions are calculated by summing up the respective con-

tributions of $CO_2$ amount fractions for fossil fuel combustion, terrestrial biospheric activities, and cement production multiplied by the $-OR$ values of $-1.4$, $-1.1$, and 0. Here the 1.4 and 1.1 are typical ORs for fossil fuel combustion and terrestrial biospheric activities, respectively. For comparison, we also calculate ER values for the $O_2$ and $CO_2$ amount fractions simulated without including the contribution of cement production. In this regard, it should be noted that Faassen et al. (2023) carried out continuous observations of $\delta(O_2/N_2)$ and the $CO_2$ amount fraction at a forest site in Finland, and they found higher ER (referred to as "ER$_{atmos}$" in their study) than 2.0 during the morning transition for the average diurnal cycle in summer. Such high ER cannot be obtained from summing up the contributions of fossil fuel combustion and terrestrial biospheric activities at the surface, and therefore they suggested the ER signal not only represents the diurnal cycle of the forest exchange but also includes other factors, including entrainment of air masses in the atmospheric boundary layer before midday, with different thermodynamic and atmospheric composition characteristics. Considering their results, we examined average diurnal cycles of $\delta(O_2/N_2)$ and the $CO_2$ amount fraction at RYO in October 2017 and August 2018 (Fig. A1a–d in Appendix A). We found the ER values are close to 1 throughout the day both for the observed and simulated diurnal cycles. Therefore, we consider the entrainment of air masses does not change the ER at RYO substantially, and the atmospheric transport processes in the AIST-MM are appropriate for comparing the observational results in the present study. The $CO_2$ amount fractions for fossil fuel combustion, terrestrial biospheric activities, and cement production calculated by the AIST-MM are available in the Supplement.

## 2.3 Extraction of a cement signal from the observed data

We extract signals of cement production based on the simultaneous measurements of $\delta(O_2/N_2)$ and $CO_2$ amount fractions. For this purpose, we use $y(CO_2{}^*)$ as an indicator:

$$y\left(CO_2{}^*\right) = y(CO_2) + \frac{X(O_2)}{\alpha_{B+F}}\delta(O_2/N_2), \quad (3)$$

where $X(O_2)$ $(= 0.2094)$ is the fraction of atmospheric $O_2$, and $\alpha_{B+F}$ is the expected ER for terrestrial biospheric activities and fossil fuel combustion. The $y(CO_2{}^*)$ is closely related to atmospheric potential oxygen $(\delta(APO))$, which is conserved for terrestrial biospheric activities (Stephens et al., 1998). Here, $y$ stands for the dry amount fraction of gas, as recommended by the IUPAC Green Book (Cohen et al., 2007). In our previous study, we calculated $\delta(APO)$ as

$$\delta(APO) = \delta(O_2/N_2) + \frac{\alpha_B}{X(O_2)}y(CO_2) - 2000 \times 10^{-6}, \quad (4)$$

where 2000 is an arbitrary reference (Ishidoya et al., 2022). For $\alpha_{B+F}$ values, we use monthly average ER values calcu-

lated from the simulated $O_2$ and $CO_2$ values without considering the contribution of cement production (dotted black line in Fig. 5, bottom, discussed below). If there are no substantial contributions from air–sea $O_2$ and $CO_2$ exchanges, then $y(CO_2{}^*)$ indicates the change in the atmospheric $CO_2$ amount fraction due only to cement production. No air–sea exchanges can be assumed if the wind field, surface ocean biological production, and ocean temperature are constant throughout the month. In fact, day-to-day variations in $\delta(O_2/N_2)$ due to the contribution of oceanic signal cannot be ignorable within a month as reported in previous studies (e.g., Goto et al., 2017). However, as shown in Figs. 5 and 6 here, variations in the $CO_2$ amount fraction due to cement production occurred over periods of less than 1 d. Taking these findings into consideration, we derived the baseline variation in $y(CO_2{}^*)$, which does not include a substantial contribution from cement production, as follows. First, we calculated the standard deviation $(1\sigma)$ of each $y(CO_2{}^*)$ value from the 24 h running means of $y(CO_2{}^*)$. Then, we removed $y(CO_2{}^*)$ values greater than the 24 h running mean of $y(CO_2{}^*) + 1\sigma$ from the analysis. Finally, we recalculated the 24 h running means by using the residual $y(CO_2{}^*)$ values, and regarded them as the baseline variation. Accordingly, the $y(CO_2{}^*)$ anomaly obtained by subtracting the baseline variation from each $y(CO_2{}^*)$ value is considered to indicate $CO_2$ changes due mainly to the contribution of the cement production.

## 3 Results and discussion

From August 2017 to November 2018, $\delta(O_2/N_2)$ and $CO_2$ amount fractions observed at RYO varied cyclically in opposite phase to each other on timescales from several hours to seasons (Fig. 2); however, variations in $CO_2$ and CO amount fractions were roughly in phase. The opposite-phase variations of $\delta(O_2/N_2)$ and $CO_2$ amount fractions were driven by fossil fuel combustion and terrestrial biospheric activities. By contrast, the atmospheric $O_2$ variation $(\mu mol\,mol^{-1})$ due to the air–sea exchange of $O_2$ is much larger than that of $CO_2$ on timescales shorter than 1 year because of the difference in their equilibration times between the atmosphere and the surface ocean: the equilibration time for $O_2$ is about 1 month and for $CO_2$ it is about 1 year because of the carbonate dissociation effect on the air–sea exchange of $CO_2$ (Keeling et al., 1993). The in-phase variations of the $CO_2$ and CO amount fractions were also driven by fossil fuel combustion and biomass burning. CO : $CO_2$ ratios for fossil fuel combustion and biomass burning reported by previous studies are about 0.01–0.04 and $> 0.1$, respectively (e.g., Nara et al., 2011; Tohjima et al., 2014; Niwa et al., 2014). The short-term (several hours to several days) variations in CO : $CO_2$ ratios were about 0.01 from late autumn to early spring, but they were much smaller in summer (Fig. 2). These results suggest, therefore, that the short-term variations in $\delta(O_2/N_2)$

and $CO_2$ amount fractions were driven mainly by fossil fuel combustion in winter and mainly by terrestrial biospheric activities in summer. Over 1 year of measurements CO amount fractions also showed a seasonal cycle with a summertime minimum that is attributed to the air mass around Japan: in winter the air mass is of continental origin and in summer it is of marine origin.

In this study, we focused on the short-term variations in $\delta(O_2/N_2)$ and the $CO_2$ and CO amount fractions (Fig. 2) to extract local effects of cement production. Therefore, we subtracted 1-week rolling average values of $\delta(O_2/N_2)$ and the $CO_2$ and CO amount fractions from the observed values to exclude their baseline variations, and examined the relationships among the residuals ($\Delta y(O_2)$, $\Delta y(CO_2)$, and $\Delta y(CO)$; Fig. 3a). Here, $\Delta y(O_2)$ is the equivalent value in $\mu mol\,mol^{-1}$ converted from $\delta(O_2/N_2)$. We also plotted the ER values calculated by least-squares fitting of regression lines to the observed $\Delta y(O_2)$ and $\Delta y(CO_2)$ values during successive 24 h periods in Fig. 3b. As seen in the figure, both ER values higher and lower than 1.1 were observed throughout the observation periods. When the terrestrial biosphere emits $CO_2$ to the atmosphere, i.e., the respiration signal is larger than the photosynthesis signal, ER values ranging from 1.05 to 2.00 are expected from combination fluxes of terrestrial biospheric activities, gas, liquid, and solid fuels combustion. Similar ER values have been observed at other Japanese sites (e.g., Minejima et al., 2012; Goto et al., 2013; Ishidoya et al., 2020).

On the other hand, when the photosynthesis signal is larger than the respiration signal, ER values for the combination fluxes could be variable and potentially even lower than 1.05. Therefore, we consider that the observed low ER values with high $\Delta y(CO)$ and $\Delta y(CO_2)$ are attributed to substantial $CO_2$ flux from cement production, of which the ER value is 0, rather than the photosynthesis signal. These characteristics can be seen from the typical ER, $\Delta y(CO)$, and $\Delta y(CO_2)$ in August 2018 plotted in Fig. 3c. Therefore, it is considered that air mass having ER values lower than 1.05 and $\Delta y(CO)$ and $\Delta y(CO_2)$ higher than 0 simultaneously indicates $CO_2$ flux from cement production mixes with the surrounding air that has already been influenced by terrestrial biospheric activities or fossil fuels combustion. Similar characteristic relationships have previously been observed only in artificial $CO_2$ release experiments of which the OR value is 0, such as those described by van Leeuwen and Meijer (2015) and Pak et al. (2016). Therefore, we used the AIST-MM model to calculate atmospheric $CO_2$ amount fractions, with or without taking into account the $CO_2$ flux from the cement plant near RYO, and to convert the calculated $CO_2$ amount fractions to $O_2$ amount fractions using the respective OR values of fossil fuels and terrestrial biospheric activities. Then we compared the observed and simulated ER values. Figure 4 shows examples of the performance of the AIST-MM in the present calculation. Figure 4a shows the monthly average of hourly $CO_2$ amount fractions is slightly overestimated at night and

underestimated in the daytime except for February; however, the absolute value of the difference is less than $2\,\mu mol\,mol^{-1}$ in most case. Figure 4b is a scatter plot of the difference from $391.14\,\mu mol\,mol^{-1}$ (the minimum concentration of observed $CO_2$ in the 7 months) between the calculated and observed concentration for all the hourly data in the 7 months. The FAC2 (fraction of calculations within a factor 2 of observations) is 0.976, where the model acceptance criterion of FAC2 is greater than 0.5 (Hanna and Chang, 2012), and Pearson's correlation coefficient is 0.69. The discrepancies between the observed and simulated values can be attributed to the limited resolution of the model in the complex terrain, or to problems in the parameterization of transport processes or in the $CO_2$ sources/sinks incorporated into the AIST-MM.

In October 2017, short-term variations in observed $CO_2$ and $\delta(O_2/N_2)$ were opposite in phase, and the amplitudes (in $\mu mol\,mol^{-1}$) of some $CO_2$ variations were larger than those of the corresponding $\delta(O_2/N_2)$ variations (Fig. 5). If the short-term variations were driven by terrestrial biospheric activities and the consumption of gas, liquid, and solid fuels, then the amplitudes of $CO_2$ should be smaller than those of the $\delta(O_2/N_2)$. Therefore, this result suggests an effect of cement production is superimposed on fossil fuel combustion and/or terrestrial biospheric activities. Similar characteristic variations suggesting a cement production effect were also seen in the observations made at RYO in November 2017 and in January, February, April, May, and August 2018 as presented in Appendix B. The simulated $CO_2$ amount fraction, calculated from the sources and sinks in east Japan with no background amount fraction by the AIST-MM, is also shown in Fig. 5. The contribution of the $CO_2$ amount fraction for the three components (cement production, terrestrial biospheric activities, and fossil fuel consumption other than cement production) is also shown in Fig. 5. The results demonstrate that cement production contributed substantially to the simulated $CO_2$ amount fraction. We examined the effect of cement production on ER values by calculating ER values by fitting regression lines to the observed and simulated $O_2$ and $CO_2$ amount fractions during successive 24 h periods (Fig. 5, bottom). Both the observed ER values and those simulated are frequently lower than 1.1, while the ER values simulated without including cement production show lower values than 1.1 occasionally (Figs. 5 and B1a–f in Appendix B). Therefore, $CO_2$ emissions from the cement plant must be incorporated into the transport model to reproduce the detailed variations in atmospheric $O_2$ and $CO_2$ amount fractions at RYO.

Next, we extracted signals of cement production based on $y(CO_2*)$ calculated from the simultaneous measurements of $\delta(O_2/N_2)$ and $CO_2$ amount fractions (see details in Sect. 2.3). In October 2017, the $y(CO_2*)$ and CO amount fraction maxima at RYO appeared at the same time that the wind was blowing from the northwest (most frequently over the range of 270–300°) (https://www.data.jma.go.jp/env/data/report/data/download/atm_bg_e.html, last access: 24 August 2021) (Fig. 6). This result suggests that the

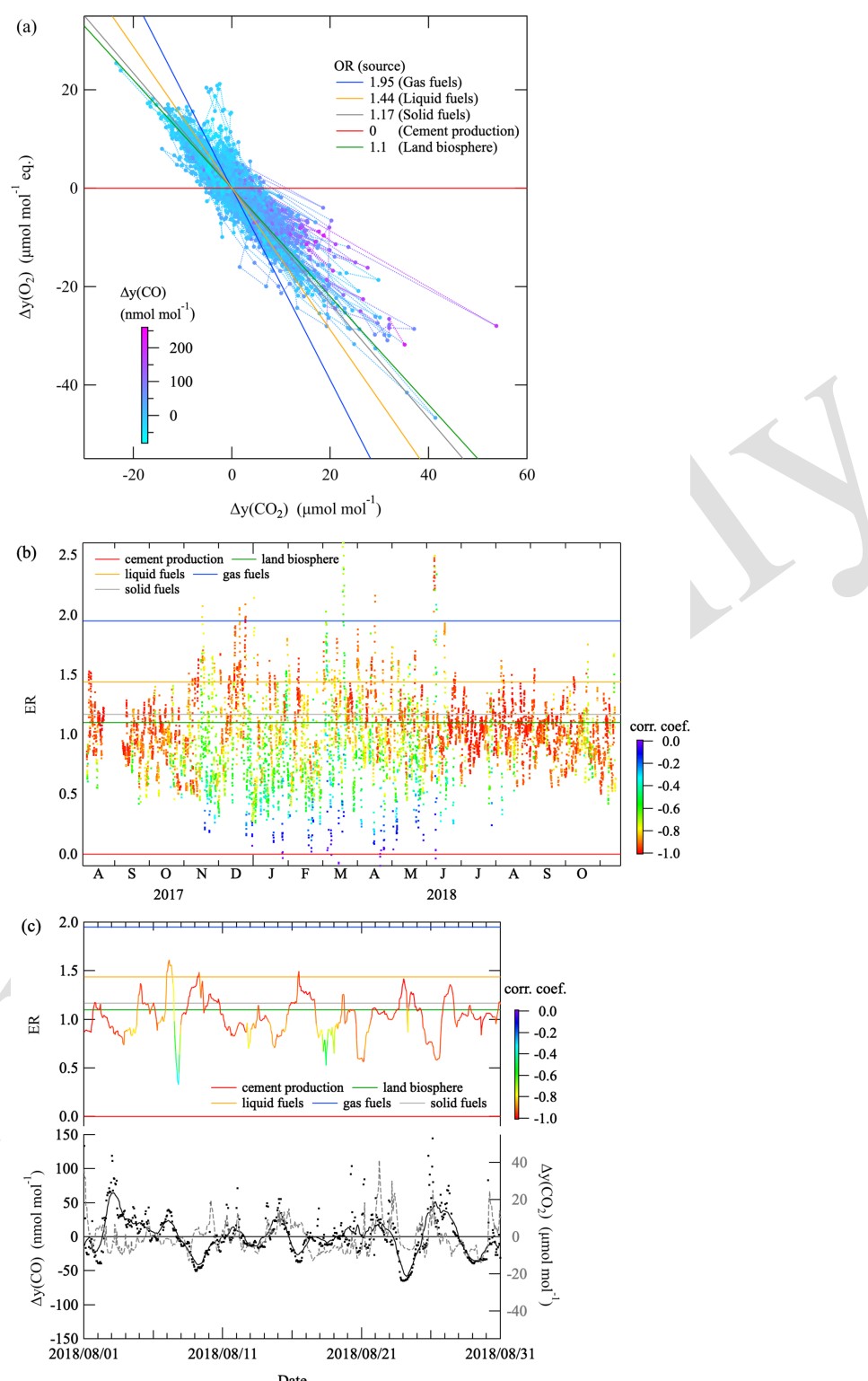

**Figure 3. (a)** Relationship between $\Delta y(O_2)$ and $\Delta y(CO_2)$ at RYO from August 2017 to November 2018. $\Delta y(O_2)$, $\Delta y(CO_2)$, and $\Delta y(CO)$ were calculated by subtracting the 1-week mean values of $\delta(O_2/N_2)$, $CO_2$, and CO amount fractions from their observed values; then $\Delta\delta(O_2/N_2)$ values were converted to the equivalent $\Delta y(O_2)$. $\Delta y(CO)$ values are shown by the color scale. The plotted ER values are from Keeling (1988) and Severinghaus (1995). **(b)** ER values calculated by least-squares fitting of regression lines to the observed $\Delta y(O_2)$ and $\Delta y(CO_2)$ values shown in **(a)** during successive 24 h periods (before and after 12 h of each point) throughout the observation period. **(c)** Same ER as in **(b)** but for August 2018. $\Delta y(CO)$ (black dots) and its 24 h averages (solid black line), and $\Delta y(CO_2)$ (dashed gray line) are also shown.

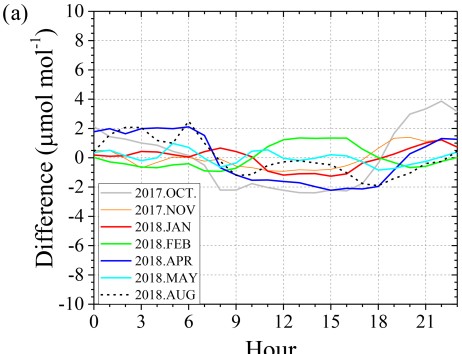

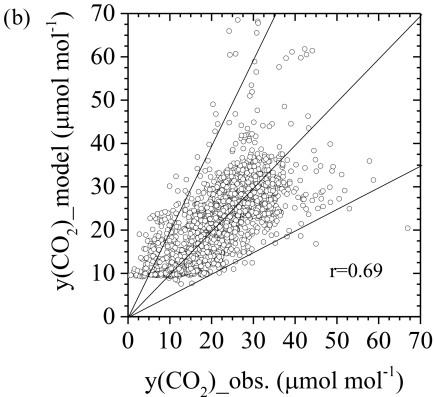

**Figure 4. (a)** Difference of monthly average of hourly amount fraction of $CO_2$ between calculated and observed concentration at RYO. **(b)** Scatter plot between observed and calculated $CO_2$ amount fraction deviation for all the hourly data of 7 months at RYO. $391.14\,\mu\text{mol}\,\text{mol}^{-1}$ (the minimum value of observed $CO_2$ amount fraction in 7 months) was subtracted from both of the data groups. Straight lines indicate the range of FAC2.

short-term variations in $y(CO_2{}^*)$ were driven mainly by air masses transported from the cement plant, which is about 6 km northwest of RYO. These findings also indicate that it is possible to extract $CO_2$ amount fraction data from background air at RYO by selecting observed ER and CO amount fraction data. We have confirmed that the present method of JMA used to select background air for the data posted on WDCGG is sufficient to exclude the effect of cement production; nevertheless, the use of ER may provide an additional constraint. Note that CO is emitted during fossil fuel combustion at the cement plant to supply electricity and heat for cement production. This means $CO_2$ is presumably released as well, so that the overall ER for the $CO_2$ emitted from cement plant (cement production + fossil fuel combustion) would not be 0.

To examine the consistency between the observed $y(CO_2{}^*)$ and simulated $CO_2$ emissions from the cement plant, we compared 5 h means of $y(CO_2{}^*)$ anomalies with changes in the $CO_2$ amount fraction due to the contribu-

tion of cement production as simulated by the AIST-MM (hereafter referred to as "$y(CO_2, \text{cement})$") (Fig. 6, bottom). The result shows that variations in the $y(CO_2{}^*)$ anomaly and $y(CO_2, \text{cement})$ are of the same order of magnitude, although they do not necessarily occur simultaneously. This result suggests that we succeeded in using $y(CO_2{}^*)$ to detect a signal of $CO_2$ emissions owing to the cement production, and that this signal can be used to validate a fine-scale atmospheric transport model. In this context, van Leeuwen and Meijer (2015) suggested that a $CO_2$ leak of $10^3\,\text{t}\,\text{a}^{-1}$ is detectable at a location up to 500 m away from the leak point based on their observations of atmospheric $O_2$ and $CO_2$ amount fractions. If this relationship follows an inverse square law, a $CO_2$ leak of $1.44 \times 10^5\,\text{t}\,\text{a}^{-1}$ should be detectable at locations up to 6 km from the leak point. Therefore, about $10^6\,\text{t}\,\text{a}^{-1}$ of the $CO_2$ emissions from the cement plant in this study, calculated with Eq. (2), is large enough to be detected at RYO. Features during November 2017, January, February, April, May, and August 2018 were similar (Fig. B2a–f in Appendix B), although the short-term variations in $y(CO_2{}^*)$ in May 2018 (Fig. B2e) were noisier than in the other months, probably because of an effect of short-term variations in the air–sea $O_2$ flux due to high primary production during the spring bloom in the nearby coastal ocean (e.g., Yamagishi et al., 2008).

The monthly mean $y(CO_2{}^*)$ anomalies shown in Fig. 7 were calculated using the ER ($\alpha_{B+F}$) value calculated by the AIST-MM for terrestrial biospheric activities and fossil fuel consumption excluding cement production. In Fig. 7, these $y(CO_2{}^*)$ anomaly values as well as those calculated using $\alpha_{B+F}$ values of 1.4 and 1.1 are compared with monthly mean $y(CO_2, \text{cement})$ values. The monthly mean $y(CO_2{}^*)$ anomalies were generally consistent with the monthly mean $y(CO_2, \text{cement})$ values from October, November, February, and April, while those were smaller in January and larger in May and August. The discrepancy between the monthly mean $y(CO_2{}^*)$ anomaly and $y(CO_2, \text{cement})$ is not explained by month-to-month changes in the cement production, since the production of clinker at the cement plant for each month was not markedly different from each other (Taiheiyo Cement Co., personal communication, 2022). We also confirmed that monthly mean $y(CO_2, \text{cement})$ values were related to the occurrence of northwesterly winds (i.e., wind blowing from the cement plant). However, the average wind direction simulated by the AIST-MM when high $y(CO_2, \text{cement})$ values appeared (around 300°) was slightly but systematically different from that for observed wind direction (around 270°) (Fig. B3a and b in Appendix B). This discrepancy is probably due to the underestimation of the altitude of Ryori ridge, which is located between the cement plant and the RYO site. Such an underestimation makes it easy to transport the $CO_2$ emitted from the cement plant directly to RYO over the ridge since the cement plant is located around 300° from the RYO site. This is also consistent with the fact that the larger monthly mean $y(CO_2, \text{cement})$ than the monthly mean $y(CO_2{}^*)$ anomalies are found in Jan-

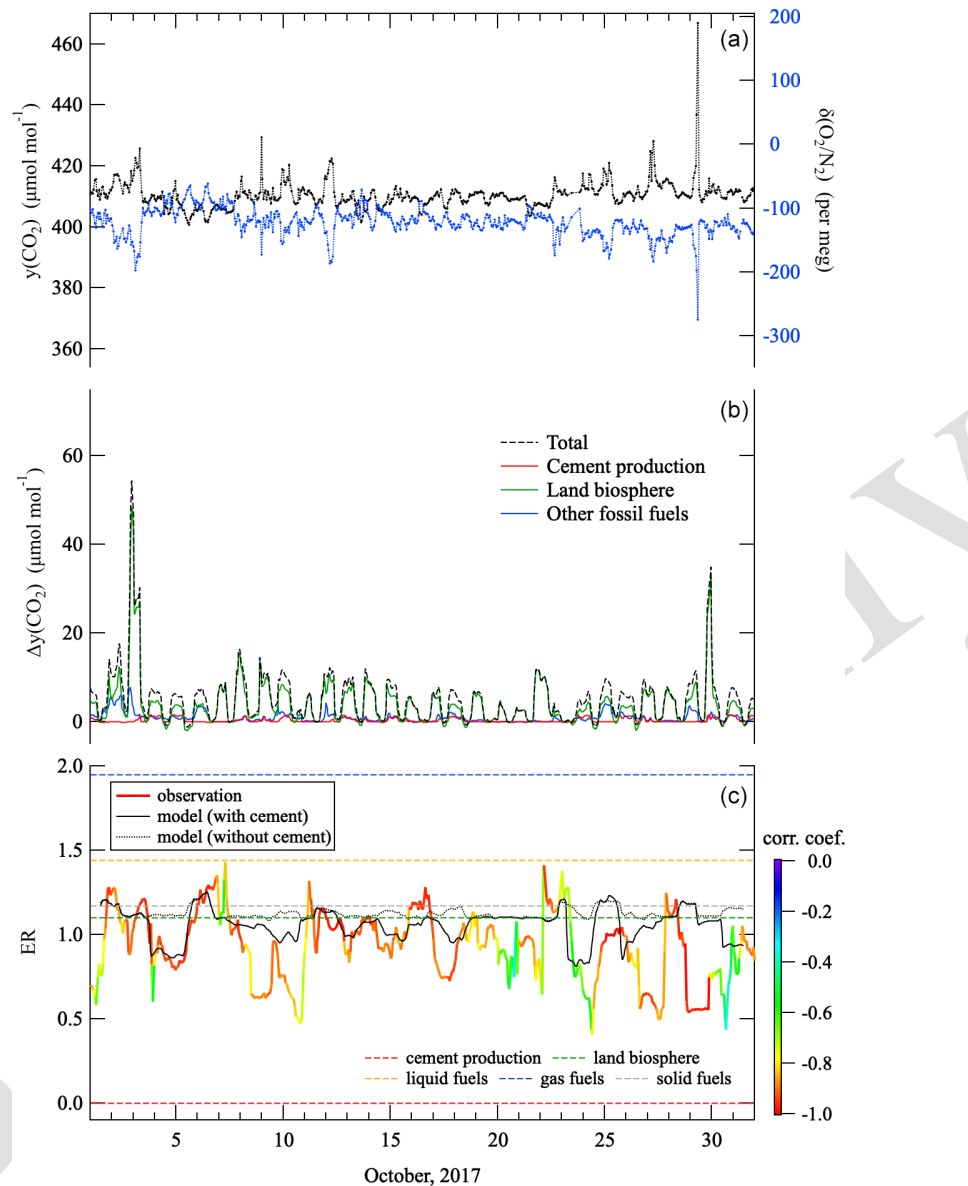

**Figure 5. (a)** Variations in CO$_2$ amount fractions and $\delta$(O$_2$/N$_2$) observed at RYO in October 2017. **(b)** Variations in the total CO$_2$ amount fraction simulated by the AIST-MM (dashed black line, see text), and the contributions of CO$_2$ amount fraction for cement production (solid red line), terrestrial biospheric activities (solid green line), and fossil fuel consumption other than cement production (solid blue line). The simulated CO$_2$ amount fractions were calculated from the sources and sinks in east Japan with no background amount fraction, i.e., $\Delta$ denotes deviations from the background amount fraction. **(c)** Variations in ER calculated by least-squares fitting of regression lines to the observed $\delta$(O$_2$/N$_2$) and CO$_2$ values during successive 24 h periods (thick colored line, where the line color indicates the value of the correlation coefficient). The corresponding ER values calculated from the simulated O$_2$ and CO$_2$ amount fractions by the AIST-MM with and without considering the amount fraction of cement production are shown by solid and dotted black lines, respectively. Dashed horizontal lines show the expected OR values for the consumption of gas, liquid, and solid fuels (Keeling, 1988); terrestrial biospheric activities (Severinghaus, 1995); and cement production.

uary and February when prevailing wind direction is north-westerly. The complex terrain around RYO such as the Ry-ori ridge would also contribute to the discrepancy between the monthly mean $y$(CO$_2{}^*$) anomaly and $y$(CO$_2$, cement) in May and August at least partly. In May, it is considered that an effect of the oceanic O$_2$ flux on $y$(CO$_2{}^*$) anomaly is also

substantial, since we can distinguish short-term variations in $\delta$(O$_2$/N$_2$) without simultaneous changes in the CO$_2$ amount fraction (Fig. B1e).

It was also found from Fig. 7 that the monthly mean $y$(CO$_2{}^*$) anomaly did not depend on the $\alpha_{B+F}$ value used to calculate $y$(CO$_2{}^*$), except August 2018. In addition, the

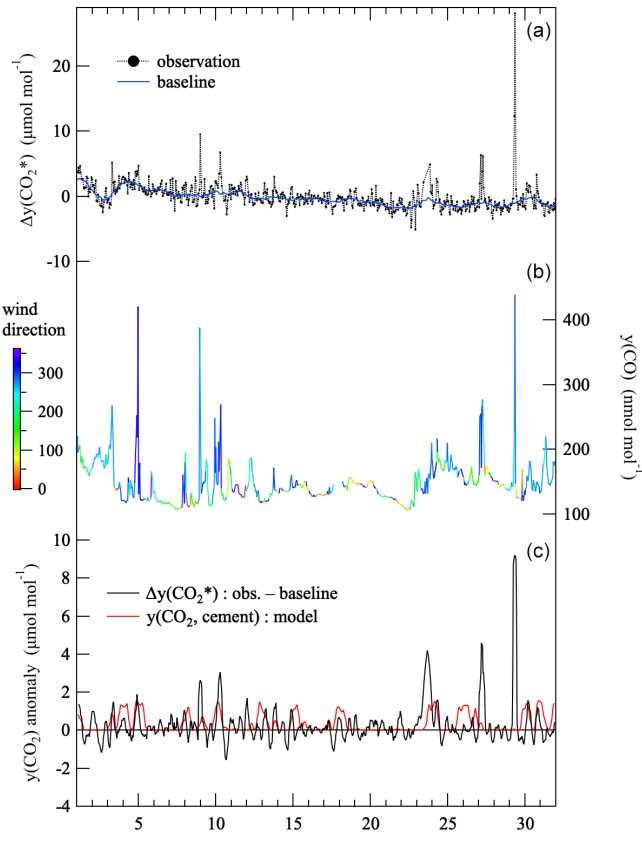

**Figure 6. (a)** Variations in $\Delta y(CO_2{}^*)$ calculated from the observed $CO_2$ amount fractions and $\delta(O_2/N_2)$ (filled black circles) in October 2017, and the baseline variation (solid blue line). $\Delta$ denotes deviations from their monthly mean values. See text for the definition of $y(CO_2{}^*)$ and the method used to obtain the baseline variation. **(b)** Variations in CO amount fractions in October 2017 and the simultaneously observed wind direction (in degrees). **(c)** 5 h-average $\Delta y(CO_2{}^*)$ anomalies from the $\Delta y(CO_2{}^*)$ baseline variation and the corresponding variation in the $CO_2$ amount fraction due only to cement production ($y(CO_2, cement)$) simulated by the AIST-MM (same as the red line in the middle part of Fig. 4).

average monthly mean $y(CO_2{}^*)$ anomaly values and the average $y(CO_2, cement)$ during the 7 months (right-hand side of Fig. 7) agreed within their monthly variabilities. These results suggest that it is not necessary to use the $\alpha_{B+F}$ value simulated by the AIST-MM to estimate the contribution of cement production to the atmospheric $CO_2$ amount fraction at RYO; rather, it can be estimated from only the observed $y(CO_2{}^*)$ by assuming an $\alpha_{B+F}$ value of 1.1 or 1.4. This is also applicable on shorter timescales (Fig. B4a and b in Appendix B). Therefore, we can derive the observed $y(CO_2{}^*)$ at RYO without using any simulated value by an atmospheric transport model, and the observed $y(CO_2{}^*)$ can be used to validate hourly to annual average $CO_2$ fluxes from cement production simulated by a fine-scale atmospheric transport model. It should also be noted that we did not use CO amount

fraction for the calculation of $y(CO_2{}^*)$. This is an important advantage to apply $y(CO_2{}^*)$ to detect $CO_2$ capture and/or $CO_2$ leak which does not emit CO.

$y(CO_2{}^*)$ is expected to be an indicator for detecting the signal of $CO_2$ capture from flue gas at the cement plant. At a cement plant, $CO_2$ is removed from flue gas without any $O_2$ changes. Therefore, if the $CO_2$ emitted during cement production, which is about $10^6\,t\,a^{-1}$ at this plant, is removed from the flue gas, then the 7-month mean $y(CO_2{}^*)$ anomaly would change from 0.4 to 0 $\mu$mol mol$^{-1}$. Thus, a cement plant can be a useful site not only for demonstrating carbon capture from flue gas but also for monitoring its efficiency based on combined measurements of $\delta(O_2/N_2)$ and $CO_2$. In addition, during the future operation of a large-scale DAC plant, a negative annual mean $y(CO_2{}^*)$ anomaly value should be observed because a DAC plant removes $CO_2$ from the atmosphere without emitting $O_2$ to the atmosphere.

## 4  Conclusions

We analyzed atmospheric $\delta(O_2/N_2)$ and $CO_2$ and CO amount fraction data observed continuously at RYO to extract a $CO_2$ emissions signal from a cement plant located about 6 km northwest of RYO. The observed $\delta(O_2/N_2)$ and $CO_2$ amount fractions varied cyclically in opposite phase to each other on timescales from several hours to seasons. From the CO : $CO_2$ ratios, the short-term variations in $\delta(O_2/N_2)$ and $CO_2$ amount fraction were inferred to be driven mainly by fossil fuel combustion in winter and by terrestrial biospheric activities in summer. We found that an ER lower than 1.1 was frequently associated with short-term variations, especially when the CO amount fraction was high; this result suggests a substantial effect of cement production, which has an ER of 0. We compared observed $CO_2$ amount fractions with those simulated by the AIST-MM for October and November 2017 and January, February, April, May, and August 2018. FAC2 for the data throughout the observation period was 0.976, which was greater than the model acceptance criterion of 0.5. Therefore, the AIST-MM-reproduced general characteristics of the observed $CO_2$ amount fraction were reproduced by the AIST-MM.

We calculated the simulated ER values by using simulated $\delta(O_2/N_2)$ values obtained from simulated $CO_2$ amount fractions and OR values of 1.1, 1.4, and 0 for terrestrial biospheric activities, fossil fuel combustion, and cement production, respectively. As in the observations, simulated ER values lower than 1.1 were frequently associated with short-term variations. $y(CO_2{}^*)$ was calculated from the observed $\delta(O_2/N_2)$ and $CO_2$ amount fractions and the simulated $\alpha_{B+F}$ to extract the cement production signal. Variations in the $y(CO_2{}^*)$ anomaly relative to baseline values were generally of the same order of magnitude as the $CO_2$ amount fraction changes due to the contribution of cement production simulated by the AIST-MM ($y(CO_2, cement)$). The monthly mean

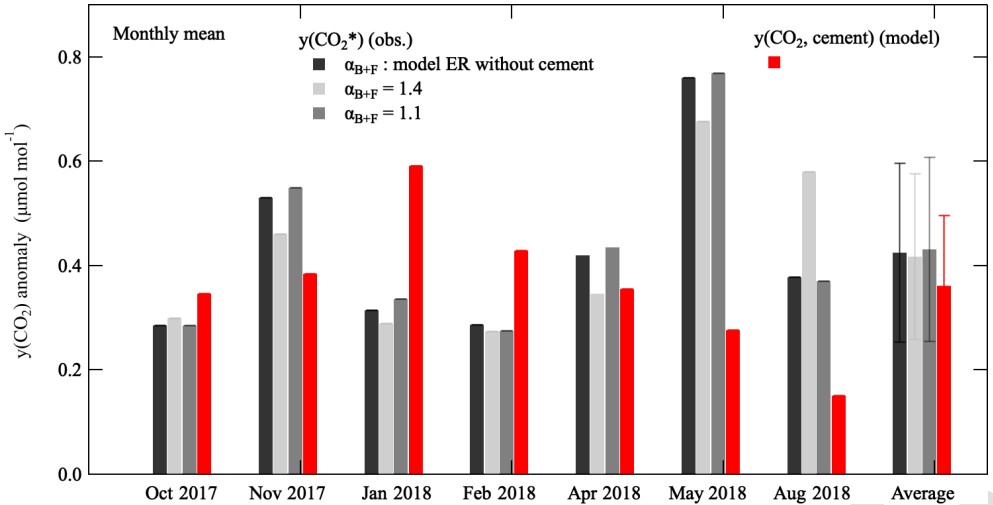

**Figure 7.** Monthly means of $y(CO_2^*)$ anomalies, obtained using model-simulated $\alpha_{B+F}$ values (as in Figs. 5 and B1a–f) and $\alpha_{B+F}$ values of 1.4 and 1.1, and $y(CO_2, \text{cement})$. The monthly mean values averaged over the 5 months are shown at the right. Error bars indicate monthly variability ($\pm 1\sigma$).

$y(CO_2^*)$ anomaly averaged over the 7 months examined in this study and the 7-month average of $y(CO_2, \text{cement})$ agreed within their variabilities.

These results confirm that monthly to annual average CO$_2$ emissions from a cement plant can be detected by using $y(CO_2^*)$, and, therefore, that a cement plant will be a useful site for demonstrating and monitoring CO$_2$ capture from flue gas in the future. As a remaining topic, some of the more detailed variations in the CO$_2$ amount fractions were not reproduced by the AIST-MM. This is at least partly due to the spatial resolution of the AIST-MM which limited its ability to reproduce air transport from a point source, such as the cement plant in the present study. In the future this work could be expanded on by using a higher-resolution atmospheric transport model to improve the agreement between the observed and simulated CO$_2$ amount fractions. An additional step could be developing a more accurate method for extracting $y(CO_2^*)$ due only to cement production, especially for the period when air–sea O$_2$ flux is substantial. This would improve the estimation of the amount of CO$_2$ capture and/or CO$_2$ leak around the observation site from an inversion analysis using the higher-resolution atmospheric transport model.

## Appendix A: Additional figures to evaluate the effect of entrainment of air mass on the observed ER

As we described in Sect. 2.2, Faassen et al. (2023) found higher ER values ("ER$_{\text{atmos}}$" in their study) than 2.0 at a forest site in Finland during the morning transition for the average diurnal cycles of $\delta(O_2/N_2)$ and the CO$_2$ amount fraction in summer. On the other hand, Ishidoya et al. (2013) reported ER values ("ER$_{\text{atm}}$" in their study) close to 1 at a Japanese forest site in summer, for the average diurnal cycles throughout the day. Considering the discrepancy between these values from Faassen et al. (2023) and Ishidoya et al. (2013), we derive the average diurnal cycle of $\delta(O_2/N_2)$ and the CO$_2$ amount fraction at RYO. For this purpose, deviations of $\delta(O_2/N_2)$ and the CO$_2$ amount fraction from their 24 h mean values were calculated, and the $\Delta\delta(O_2/N_2)$ were converted to $\Delta y(O_2)$ by multiplying $X(O_2)$ ($= 0.2094$). Figure A1a–b show the average diurnal cycles of $\Delta y(O_2)$ and $\Delta y(CO_2)$ in October 2017, and their relationship. Those for August 2018 are also shown in Fig. A1c–d. As seen from the figures, the observed $\Delta y(O_2)$ took maxima in the daytime, and the ER values for the average diurnal cycles at RYO were close to 1 throughout the day. The corresponding diurnal $\Delta y(O_2)$ and $\Delta y(CO_2)$ cycles and their relationships obtained from the simulated results by the AIST-MM are also shown in Fig. A1a–d. Similar to the observations, it was found that the simulated $\Delta y(O_2)$ took maxima in the daytime and the ERs were close to 1 throughout the day. These facts indicate the observed ER at RYO can be reproduced by the AIST-MM generally, including the period during the morning transition. Therefore, an entrainment of air mass to yield high ER during the morning suggested by Faassen et al. (2023) may be a characteristic phenomenon at their observational site.

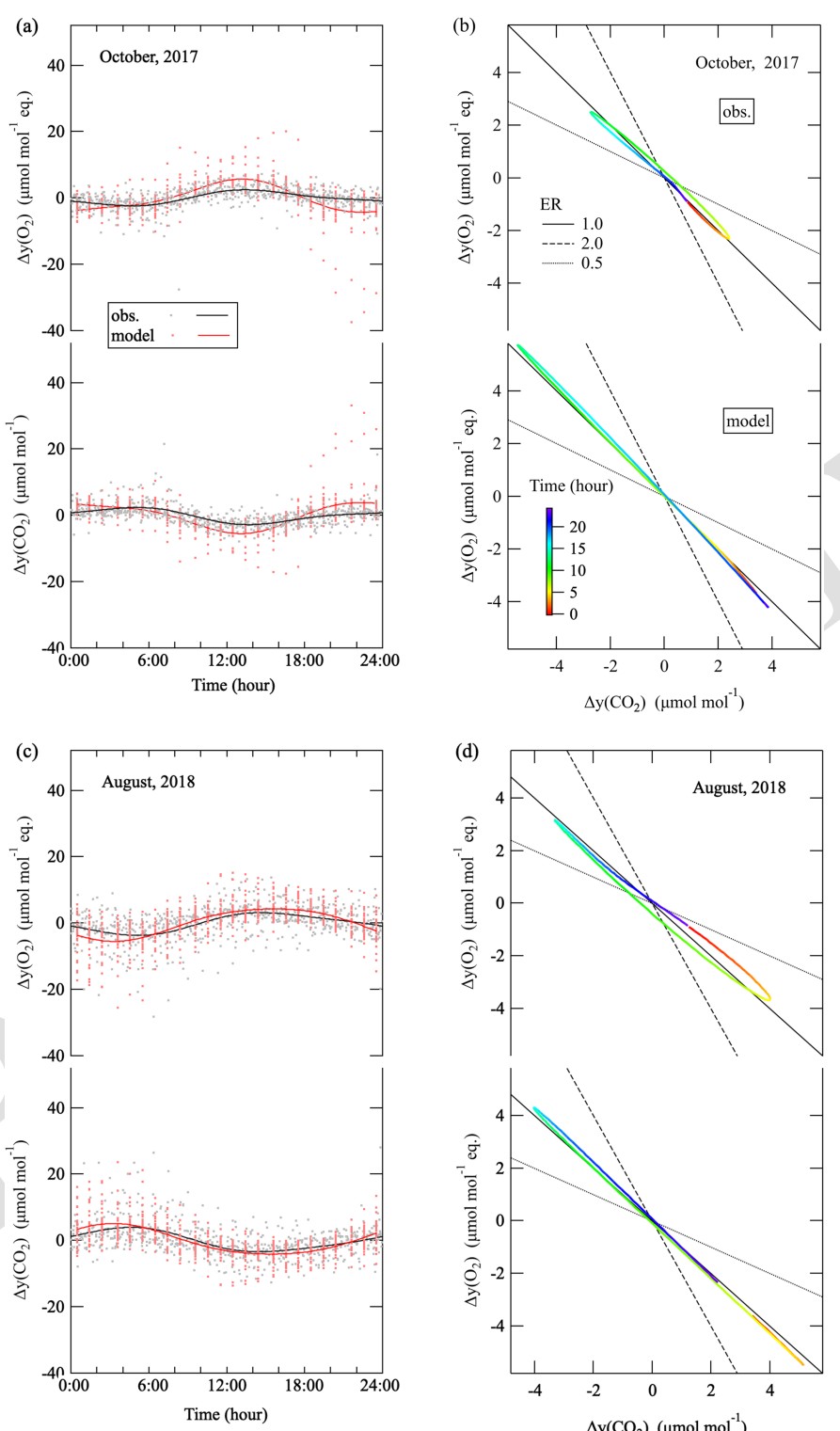

**Figure A1. (a)** Average diurnal cycles of the observed $\Delta y(O_2)$ and $\Delta y(CO_2)$ (gray dots) in October 2017. $\Delta$ denotes deviations of $\delta(O_2/N_2)$ and $y(CO_2)$ from their 24 h mean values. The $\Delta\delta(O_2/N_2)$ was converted to $\Delta y(O_2)$ by multiplying $X(O_2)$ ($= 0.2094$). Best-fit curves to the data, represented by the fundamental and its first harmonics (periods of 24 and 12 h) terms, are also shown (black lines). Those of $\Delta y(O_2)$ and $\Delta y(CO_2)$ (red dots) and best-fit curves (red lines) simulated by the AIST-MM are also shown. **(b)** Relationships between the best-fit curves of the observed (top panel) and simulated (bottom panel) $\Delta y(O_2)$ and $\Delta y(CO_2)$ shown in **(a)**. The color scale denotes the time of the day. The relationships expected from the ER of 1.0, 2.0, and 0.5 are also shown by solid, dashed, and dotted black lines, respectively. **(c)** Same as in **(a)** but for August 2018. **(d)** Same as in **(b)** but for August 2018.

## Appendix B: Additional figures to evaluate the effect of cement production on the observed and simulated CO$_2$ amount fractions

In the main text, variations in CO$_2$ amount fractions and $\delta$(O$_2$/N$_2$) observed at RYO, CO$_2$ amount fractions simulated by the AIST-MM, and ERs calculated from the observed and simulated data in October 2017 are shown in Fig. 5. We also show the corresponding figures in November 2017 and in January, February, April, May, and August 2018 in Fig. B1a, b, c, d, e, and f, respectively. Variations in $y$(CO$_2$*), CO amount fractions in October 2017, and 5 h averages of the $y$(CO$_2$*) anomalies from the $y$(CO$_2$*) baseline variation and those of $y$(CO$_2$, cement) simulated by the AIST-MM are shown in Fig. 6. We also show the corresponding figures in November 2017 and in January, February, April, May, and August 2018 in Fig. B2a, b, c, d, e, and f, respectively. General characteristics of Figs. B1a–f and B2a–f are found to be similar to those discussed in the main text for Figs. 5 and 6, respectively. However, we can distinguish short-term variations in $\delta$(O$_2$/N$_2$) without simultaneous changes in the CO$_2$ amount fraction in May 2018 (Fig. B1e), which may be attributed to substantial oceanic O$_2$ flux due to high primary production during the spring bloom.

Figure B3a shows relationships between $y$(CO$_2$*) and wind direction at RYO. Same as in Fig. B3a but for $y$(CO$_2$, cement) simulated by the AIST-MM is shown in Fig. B3b. The average wind direction when high $y$(CO$_2$, cement) values appeared is around 300°, while that for observed wind direction is around 270°. This discrepancy is probably due to insufficient spatial resolution of the AIST-MM as discussed in the main text.

Figure B4a and b show the bottom panels of Figs. 6 and A2a, respectively, but for adding the $\Delta y$(CO$_2$*) calculated by using the $\alpha_{B+F}$ values of 1.4 and 1.1. As seen from the figures, several hours to day-to-day variations in the $\Delta y$(CO$_2$*) did not change substantially depending on the $\alpha_{B+F}$ value used to calculate $y$(CO$_2$*). Therefore, the contribution of cement production to the atmospheric CO$_2$ amount fraction at RYO can be estimated from the observed $y$(CO$_2$*) by assuming an $\alpha_{B+F}$ value of 1.1 or 1.4, not only for the monthly timescale but for shorter (hourly to day-to-day) timescales.

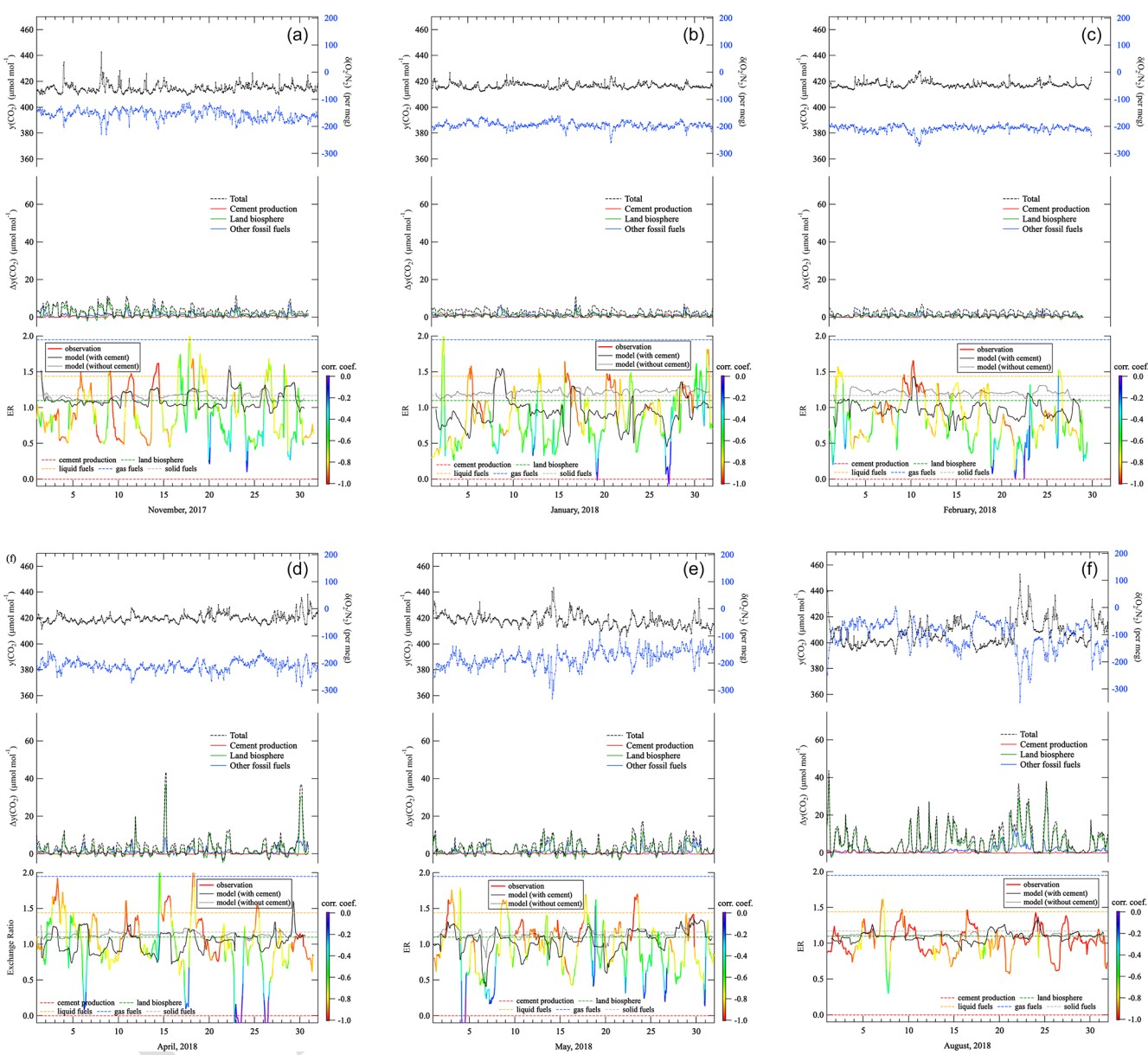

**Figure B1. (a)** Same as in Fig. 5, but for November 2017. **(b)** As in **(a)**, but for January 2018. **(c)** As in **(a)**, but for February 2018. **(d)** As in **(a)**, but for April 2018. **(e)** As in **(a)**, but for May 2018. **(f)** As in **(a)**, but for August 2018.

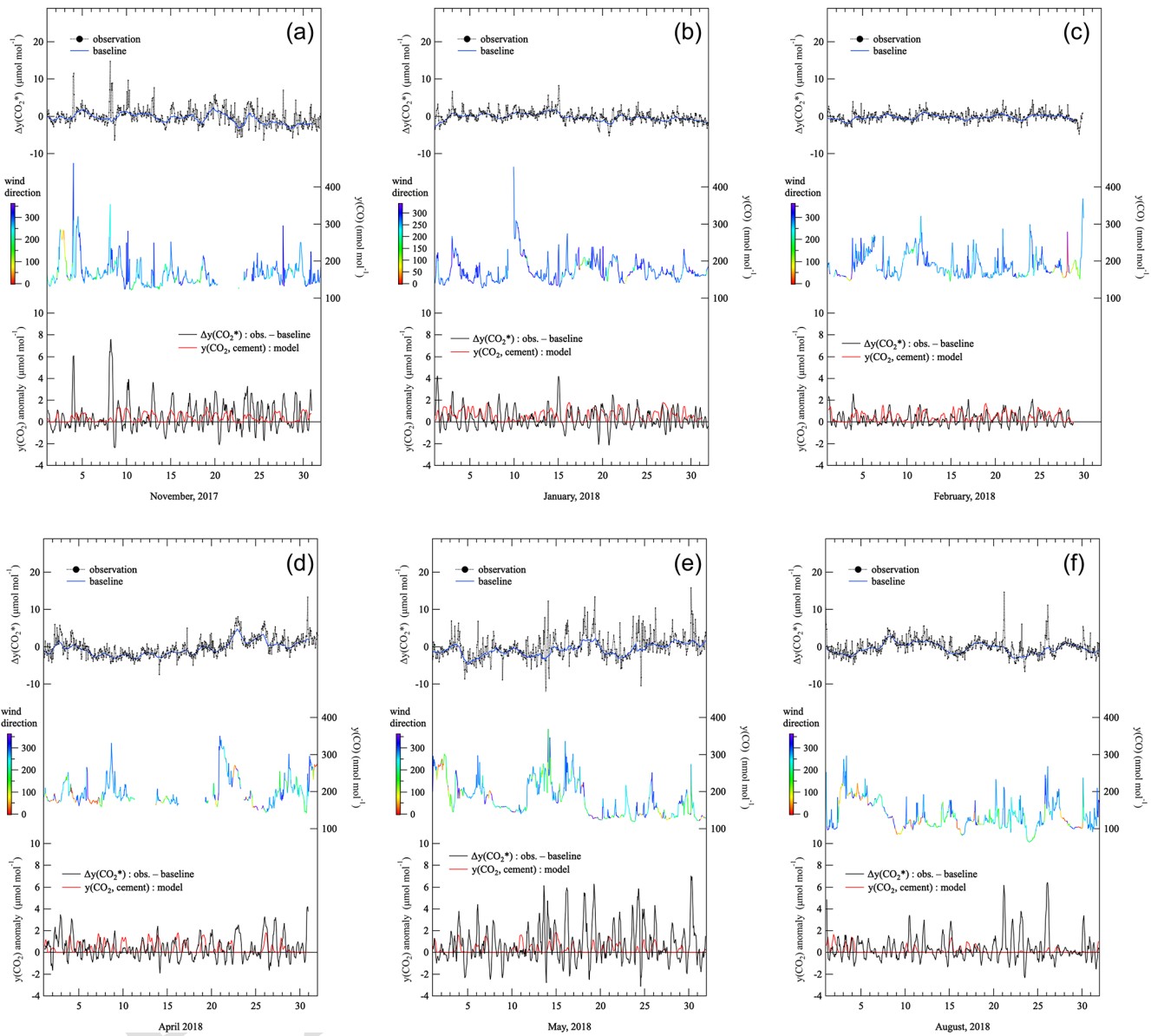

**Figure B2. (a)** Same as in Fig. 6, but for November 2017. **(b)** As in **(a)**, but for January 2018. **(c)** As in **(a)**, but for February 2018. **(d)** As in **(a)**, but for April 2018. **(e)** As in **(a)**, but for May 2018. **(f)** As in **(a)**, but for August 2018.

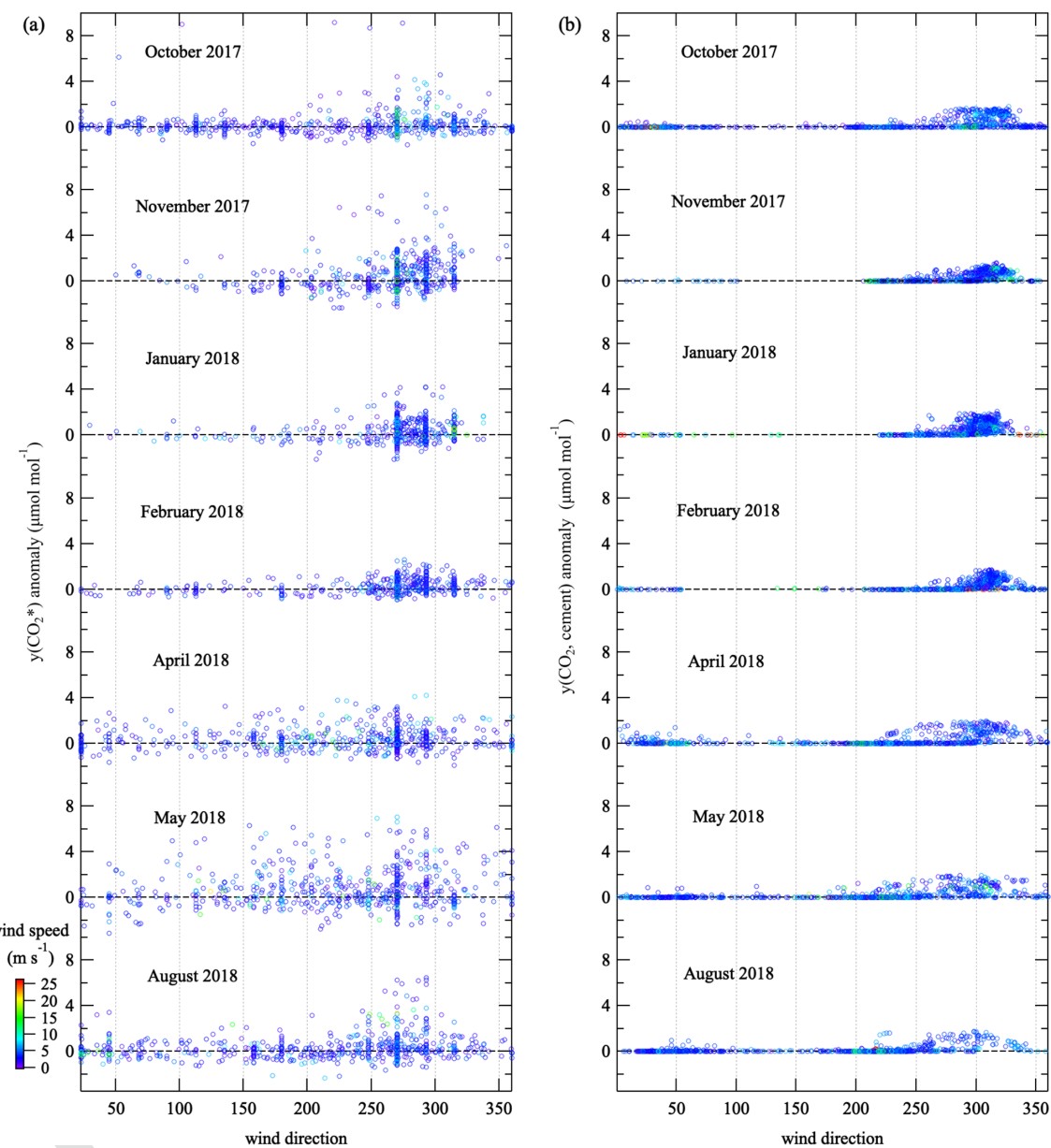

**Figure B3.** (a) Relationships between $y(CO_2{}^*)$ anomaly shown in Figs. 6 and B2 and wind direction at RYO. (b) Same as in (a) but for $y(CO_2, \text{cement})$. It is noted that the $y(CO_2{}^*)$ anomaly is 5 h average similar to Figs. 6 and B2 but the $y(CO_2, \text{cement})$ is hourly values.

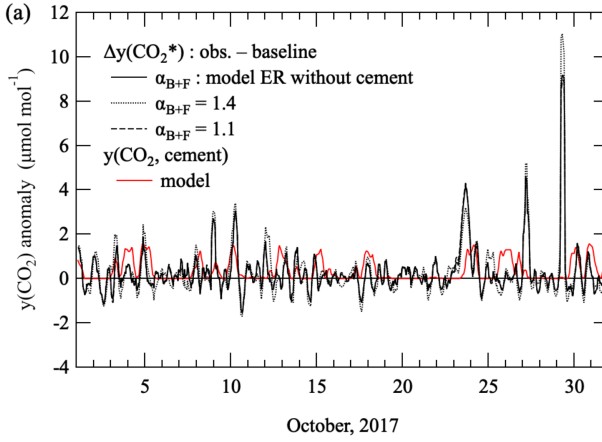

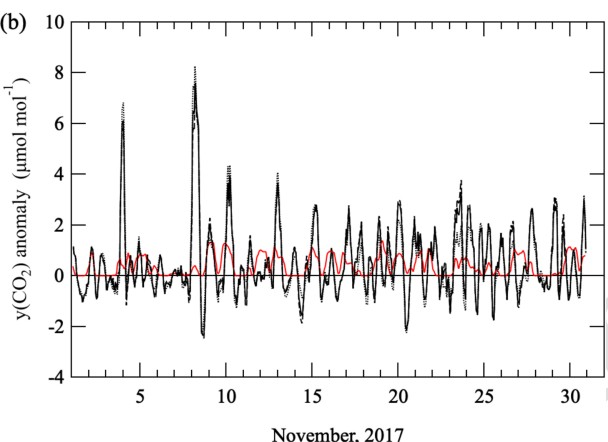

**Figure B4. (a)** Same as in the bottom panels of Fig. 6, but for $\Delta y(CO_2{}^*)$ calculated by using model-simulated $\alpha_{B+F}$ values (solid black line), and $\alpha_{B+F}$ values of 1.4 (dotted black line) and 1.1 (dashed black line). **(b)** Same as in **(a)** but for the bottom panels of Fig. B2a.

**Data availability.** The observational data of $\delta(O_2/N_2)$ and the $CO_2$ amount fraction are available through the World Data Centre for Greenhouse Gases (WDCGG) at https://gaw.kishou.go.jp (last access: 10 November 2023), and the respective DOIs are https://doi.org/10.50849/WDCGG_0006-2012-7001-01-01-9999 (Ishidoya, 2023a) and https://doi.org/10.50849/WDCGG_0006-2012-1001-01-01-9999 (Ishidoya, 2023b).

**Supplement.** The supplement related to this article is available online at: https://doi.org/10.5194/acp-24-1-2024-supplement.

**Author contributions.** SI designed the study and drafted the manuscript. Measurements of $O_2$ and $CO_2$ amount fractions were conducted by SI, KT, and KS. HK conducted the AIST-MM simulations. NA prepared the standard gas for the $O_2$ measurements. KI and HM examined the results and provided feedback on the manuscript. All authors approved the final manuscript.

**Competing interests.** The contact author has declared that none of the authors has any competing interests.

**Disclaimer.** Publisher's note: Copernicus Publications remains neutral with regard to jurisdictional claims made in the text, published maps, institutional affiliations, or any other geographical representation in this paper. While Copernicus Publications makes every effort to include appropriate place names, the final responsibility lies with the authors.

**Acknowledgements.** We acknowledge the many staff members of the Japan Meteorological Agency. We also thank Shohei Murayama at the National Institute of Advanced Industrial Science and Technology (AIST), Ryo Fujita at the Meteorological Research Institute, and JANS Co. Ltd. for supporting the research.

**Financial support.** This study was partly supported by JSPS KAKENHI (grant nos. 19H01975, 22H03739, and 22H05006) and the Global Environment Research Coordination System from the Ministry of the Environment, Japan (grant nos. METI1454 and METI1953).

**Review statement.** This paper was edited by Jan Kaiser and reviewed by two anonymous referees.

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
