# Peer review of "Measurement report: Method for evaluating CO2 emissions from a cement plant using atmospheric $\delta(O_2/N_2)$ and CO2 measurements and its implication for future detection of CO2 capture signals"

_Atmospheric Chemistry and Physics, 2022_

## Referee Comment (RC1)

**Review of: Measurement report: Method for evaluating CO2 emission from a cement plant by atmosphere O2/N2 and CO2 measurements and its applicability to the detection of CO2 capture signals**

This is an interesting paper that presents important data and should be published. I have a number of suggestions for improvement, however. Most of my comments are minor.

1. Don't think you need "Measurement report" in the title
2. The title doesn't sound quite right to me. I think "by atmosphere O2/N2 and CO2 measurements" should be "using atmosphere O2/N2 and CO2 measurements"
3. In a few places you have written "$CO_2$ emission", I think it should be "$CO_2$ emissions". Emissions are usually written as a plural.
4. The $O_2/N_2$ and $CO_2$ are referred to as "amount fractions", I think it should be either "mole fractions" or "molar fractions".
5. In some places you use "analyses" and I think it should be "analysis". If you're referring to one study, you use the word "analysis." But when you're referring to multiple studies, you use the term "analyses." Also, you use "analyzed" and I think this should be "analysed".
6. Need to be careful when using the words "significantly" & "significant" as this usually implies some sort of statistical test with a p-value. In some places maybe change it to "substantially".
7. Lines 16-18: "The simulated CO2 amount fractions were converted to O2 amount fractions by using the respective OR values for each of the incorporated CO2 fluxes, and then simulated OR values were calculated from the calculated O2 and CO2 amount fractions." Rephrase this sentence as it is difficult to follow, its sounds like you used OR values to calculate $O_2$ and then used $O_2$ to calculate OR values.
8. Line 30: Should probably also include here reference for Pickers et al., 2022, Science Advances
9. Lines 32-34: These ratios are typically very stable/ tend not to vary very much. The analysis you do later on is only possible because these ratios are so reliable so need to say so somewhere. Also, mention here that 1.4 is global average for fossil fuels, as you use the 1.4 ratio later on.
10. Lines 34-36: "Therefore, atmospheric O2 and CO2 fluxes due to terrestrial biospheric activities and fossil fuel combustion (excluding cement production) vary in opposite phase." This sentence needs to be reworded, as respiration in the terrestrial biosphere takes in O2 and releases $CO_2$ which is the same phase as fossil fuel combustion.
11. Line 39: There is a full stop at the end of the equation which I don't think should be there.
12. Line 43: I think "global fossil $CO_2$ emissions" should be changed to "global fossil fuel $CO_2$ emissions"
13. Line 70: It says that you have been continuously making measurements since 2017. That implies that the measurements are still ongoing. But if that is the case, why are you only using measurements until 19[th] November 2018? If you have more data for 2019 to 2022 you should include this, as at the moment you only have 16 months of measurements, and more measurements to base the conclusions on would make the study better.
14. Methods: In the first paragraph of the methods section, say that the measurement site is on the coast.
15. Line 86: 1/0.2094 is 4.78 to 2 decimal places and 4.8 to 1 decimal place. I'm not expecting you to change it for this article, because it will make a tiny difference, but in the future, you may want to think about whether you really want to round it to 1 decimal place. I think lots of studies instead of dividing by 4.8, will multiply by 0.2094. This is a wider problem within

the $O_2$ community, but we should probably try and have $O_2$ datasets produced using the same method, so different groups can be compared.

16. Line 96: Should probably say something like, "gaps in the data due to routine calibrations, maintenance and technical issues". I can see from Figure 2 that there is a gap at the end of August/ beginning of September 2017. Might also want to say that there are X number of data points or what percentage of the time period has data.

17. Line 97: I think you need more detail about the measurement system, instead to referring the readers to another article. Add in a sentence or two summarizing the measurement system. Then you can say "for more information see Ishidoya et al. (2017)".

18. Line 108: You talk about the $CO_2$ calibration scale but not the $O_2$ scale. In Figure 2 the $O_2$ data is 0 to -300 per meg so you can't be using the Scripps scale. Need to add information about the $O_2$ calibration scale.

19. Line 120: Should probably explain what "clinker" is.

20. Results and discussion: There is only just over 1 year of measurements so need to be careful about the wording when talking about the seasonal cycle. In order to properly investigate seasonal cycles at least a few years of data are needed. So try rephrasing to something like "over one year of measurements CO showed a seasonal cycle with" or "in 2017/2018 the seasonal cycle was", etc.

21. Lines 132-135: It says that $O_2$ exchange is faster than $CO_2$ but I'd actually put the timescales in here, $O_2$ is about a month and $CO_2$ is about a year.

22. Line 145: Change "1-week average" to "1-week rolling average"

23. Line 153: I think you need to explain more clearly what the ratios tell you. 1.05 to 2.00 indicates terrestrial or fossil fuel. Anything lower than this indicates cement as the 0 ratio mixes with the surrounding air that has already been influenced by terrestrial or fossil fuels pulling down the ratio.

24. Lines 157-160: Why did this study choose these particular 5 months to focus on? Does this mean that not every month has evidence of cement production, or these were the months where the evidence was largest? And if so why do you think this is, there was less cement production taking place at the plant then, or air was coming from the direction of the plant less often?

25. Lines 157-159: "In October 2017, short-term variations in observed CO2 and d(O2/N2) were opposite in phase, and the amplitudes of some CO2 variations were larger than those of the corresponding d(O2/N2) variations. This result suggests an effect of cement production." I think these two sentences don't join together properly. $CO_2$ & $O_2/N_2$ opposite in phase doesn't suggest cement production, $CO_2$ increasing and $O_2/N_2$ staying the same would suggest cement production.

26. Line 168: Change "land biospheric" to "terrestrial biospheric" as that is what you have used everywhere else.

27. Lines 191-192: Used "however" twice in two sentences.

28. Line 208: Change "This means CO2 presumably as well" to ""This means CO2 is presumably released as well"

29. Line 213: "$CO_{2cement}$"

30. Summary: Articles usually include something about "next steps", how the research could be developed in the future.

31. Also say something about the limitations of the study. Although the limitations can go in the results and discussion section if you think it will fit better there.

32. Line 284: In the acknowledgments change "observation" to "observations"

33. Lots of Figures: Figure 4 and Figure 5 are actually 5 figures each, (a-e) for each of the months. I think this is probably too many figures. Could you try combining them in some way, or choose an example month and move the others to the supplement.
34. In Figure 2 and the top panels of the Figure 4's $CO_2$ is in units of $\mu mol\ mol-1$. Isn't this just ppm units, that is what most people are more familiar with?
35. Figure 2 Caption:  Change "1-week average" to "1-week rolling average"
36. Figure 2 Caption: Add something about how the $CO_2$ & $O_2$ y-axes are scaled to be visually comparable or the $O_2$ y-axis is 5 times larger than the $CO_2$ y-axis or something like that.
37. Figure 2: Could you add another panel for the Oxidative Ratio. I know we can see it for some of the individual months but I'm curious to see it for the whole time period.
38. Figure 3 Caption: "Severinghous"
39. The supplement doesn't include any of the model output or the CO measurements.

---

## Author Comment (AC1)

**Responses to Referee 1**

This is an interesting paper that presents important data and should be published. I have a number of suggestions for improvement, however. Most of my comments are minor.

Thank you very much for your significant and useful comments on the paper "Measurement report: Method for evaluating $CO_2$ emissions from a cement plant using atmosphere $\delta(O_2/N_2)$ and $CO_2$ measurements and its implication for future detection of $CO_2$ capture signals" by Ishidoya et al. We have revised the manuscript, considering your comments and suggestions. Details of our revision are as follows. The line numbers denote those of the revised manuscript.

Don't think you need "Measurement report" in the title

We understand your suggestion, however, I wrote the phrase following the Editor's comment.

The title doesn't sound quite right to me. I think "by atmosphere O2/N2 and CO2 measurements" should be "using atmosphere O2/N2 and CO2 measurements"

Title: We have modified the title, as suggested.

The O2/N2 and CO2 are referred to as "amount fractions", I think it should be either "mole fractions" or "molar fractions".

We recognize "mole fractions" you suggested is more familiar with our research field, however, I have used the phrase following the Editor's comment.

In some places you use "analyses" and I think it should be "analysis". If you're referring to one study, you use the word "analysis." But when you're referring to multiple studies, you use the term "analyses." Also, you use "analyzed" and I think this should be "analysed".

We have changed "analyses" to "analysis" and "analyzed" to "analysed" throughout the paper, following your comments.

Need to be careful when using the words "significantly" & "significant" as this usually implies some sort of statistical test with a p-value. In some places maybe change it to "substantially"

We have changed "significantly/significant" to "substantially" at some places, as suggested.

Lines 16-18: "The simulated CO2 amount fractions were converted to O2 amount fractions by using the respective OR values for each of the incorporated CO2 fluxes, and then simulated OR values were calculated from the calculated O2 and CO2 amount fractions." Rephrase this sentence as it is difficult to follow, its sounds like you used OR values to calculate O2 and then used O2 to calculate OR values

Lines 16-18: The sentences have been rewritten as "The simulated $CO_2$ amount fractions were converted to $O_2$ amount fractions by using the respective ER values of 1.1, 1.4, and 0 for the terrestrial biospheric activities, fossil fuel combustion, and cement production. Thus obtained $O_2$ and $CO_2$ amount fractions changes were used to derive simulated ER for comparison with the observed ER.".

Line 30: Should probably also include here reference for Pickers et al., 2022, Science Advances
Lines 31-32: We cited Pickers et al. (2022) here, as suggested.

Lines 32-34: These ratios are typically very stable/ tend not to vary very much. The analysis you do later on is only possible because these ratios are so reliable so need to say so somewhere. Also, mention here that 1.4 is global average for fossil fuels, as you use the 1.4 ratio later on.
Lines 37-38: The sentence "The ERs are typically very stable, and the global average ER for fossil fuels is about 1.4 (e.g. Keeling and Manning, 2014)" has been added, considering your suggestion.

Lines 34-36: "Therefore, atmospheric O2 and CO2 fluxes due to terrestrial biospheric activities and fossil fuel combustion (excluding cement production) vary in opposite phase." This sentence needs to be reworded, as respiration in the terrestrial biosphere takes in O2 and releases CO2 which is the same phase as fossil fuel combustion.
Lines 34-36: The sentence has been rewritten as "Therefore, atmospheric $O_2$ amount fraction varies in opposite phase with $CO_2$ amount fraction, owing to terrestrial biospheric activities and fossil fuel combustion" to avoid confusion.

Line 39: There is a full stop at the end of the equation which I don't think should be there.
Lines 40: Since our past paper published in ACP, there are some cases we used a full stop at the end of the equation. So, we leave it as it is, but we will revise it if the editorial support team also instructs us to change the format.

Line 43: I think "global fossil CO2 emissions" should be changed to "global fossil fuel CO2 emissions".
Lines 44: The words "global fossil $CO_2$ emissions" has been changed to "global fossil fuel $CO_2$ emissions", as suggested.

Line 70: It says that you have been continuously making measurements since 2017. That implies that the measurements are still ongoing. But if that is the case, why are you only using measurements until 19th November 2018? If you have more data for 2019 to 2022 you should include this, as at the

moment you only have 16 months of measurements, and more measurements to base the conclusions on would make the study better.

As you pointed out, the measurements are still ongoing now. However, we consider the data presented in this paper are enough to discuss an effect of cement production. Also, due to our manpower constraint, we cannot calculate the atmospheric $CO_2$ amount fraction using the fine-scale 3-D atmospheric transport model (AIST-MM) longer time period than the seven months presented in the revised manuscript. Therefore, we leave the observational data period as they are in the revised manuscript.

Methods: In the first paragraph of the methods section, say that the measurement site is on the coast.

Lines 72-73: The words has been modified as "Atmospheric $\delta(O_2/N_2)$ and $CO_2$ amount fractions have been observed continuously at a coastal station Ryori (RYO: 39° 2′ N, 141° 49′ E, 260 m a.s.l.; Fig. 1)….", to say that the measurement site is on the coast.

Line 86: 1/0.2094 is 4.78 to 2 decimal places and 4.8 to 1 decimal place. I'm not expecting you to change it for this article, because it will make a tiny difference, but in the future, you may want to think about whether you really want to round it to 1 decimal place. I think lots of studies instead of dividing by 4.8, will multiply by 0.2094. This is a wider problem within the O2 community, but we should probably try and have O2 datasets produced using the same method, so different groups can be compared.

Lines 90-91: We understand your concern. In this study, we converted changes in $\delta(O_2/N_2)$ to those in $O_2$ amount fraction by multiplying by 0.2094 in actual. So, we have rewritten the sentence as "Therefore, observed relative changes in $\delta(O_2/N_2)$ were converted to those in $O_2$ amount fraction by multiplying by 0.2094 $\mu$mol mol$^{-1}$ (per meg)$^{-1}$".

Line 96: Should probably say something like, "gaps in the data due to routine calibrations, maintenance and technical issues". I can see from Figure 2 that there is a gap at the end of August/ beginning of September 2017. Might also want to say that there are X number of data points or what percentage of the time period has data.

Lines 109-111: The sentences "It should be noted that gaps in the data seen at the end of August to beginning of September 2017 are due to maintenance and technical issues other than routine calibrations described above. The number of $\delta(O_2/N_2)$ (and $CO_2$ amount fraction) data points shown in Fig. 2 is 9221" have been added, as suggested.

Line 97: I think you need more detail about the measurement system, instead to referring the readers to another article. Add in a sentence or two summarizing the measurement system. Then you can say "for more information see Ishidoya et al. (2017)".

Lines 92-108: We have added some sentences to describe the measurement procedures in more detail, as suggested.

Line 108: You talk about the CO2 calibration scale but not the O2 scale. In Figure 2 the O2 data is 0 to -300 per meg so you can't be using the Scripps scale. Need to add information about the O2 calibration scale.

Lines 92-95: The sentences have been added to add information about the $O_2$ calibration scale as "In this study, $\delta(O_2/N_2)$ of each air sample was measured with a paramagnetic analyzer using high- and low-span standard air of which $\delta(O_2/N_2)$ had been measured against our primary standard air (Cylinder No. CRC00045; AIST-scale) using a mass spectrometer (Thermo Scientific Delta-V) (Ishidoya and Murayama, 2014). The scale based on the primary standard air is our original scale, called as "EMRI/AIST scale" in Aoki et al. (2021)".

Line 120: Should probably explain what "clinker" is.

Lines 135-137: To explain what "clinker" is, the sentences have been rewritten as "The $CO_2$ emissions from the cement plant were estimated from the clinker production capacity of the Ofunato plant in 2018 (Japan Cement Association 2020). The clinker is a solid material produced in the cement manufacture as an intermediary product of Portland cement, mainly consisting of $CaO$, $SiO_2$, $Al_2O_3$ and $Fe_2O_3$".

Results and discussion: There is only just over 1 year of measurements so need to be careful about the wording when talking about the seasonal cycle. In order to properly investigate seasonal cycles at least a few years of data are needed. So try rephrasing to something like "over one year of measurements CO showed a seasonal cycle with" or "in 2017/2018 the seasonal cycle was", etc.

Lines 189-191: The sentence has been modified as "Over one year of measurements CO amount fractions also showed a seasonal cycle with a summertime minimum that is attributed to the air mass around Japan: in winter the air mass is of continental origin and in summer it is of maritime origin", following your suggestion.

Lines 132-135: It says that O2 exchange is faster than CO2 but I'd actually put the timescales in here, O2 is about a month and CO2 is about a year.

Lines 180-183: The sentence has been modified as "In contrast, the atmospheric $O_2$ variation ($\mu mol$ $mol^{-1}$) due to the air–sea exchange of $O_2$ is much larger than that of $CO_2$ on timescales shorter than 1

year because of the difference in their equilibration times between the atmosphere and the surface ocean: the equilibration time for $O_2$ is about a month and $CO_2$ is about a year because of the carbonate dissociation effect on the air–sea exchange of $CO_2$ (Keeling et al., 1993)", following your suggestion.

Line 145: Change "1-week average" to "1-week rolling average".
Line 193: The words "1-week average" have been changed to "1-week rolling average".

Line 153: I think you need to explain more clearly what the ratios tell you. 1.05 to 2.00 indicates terrestrial or fossil fuel. Anything lower than this indicates cement as the 0 ratio mixes with the surrounding air that has already been influenced by terrestrial or fossil fuels pulling down the ratio.
Lines 192-209: Considering your comments and another referee's comments, we have expanded the discussion by adding new figures (Fig. 3b, c). Please confirm the revised manuscript.

Lines 157-160: Why did this study choose these particular 5 months to focus on? Does this mean that not every month has evidence of cement production, or these were the months where the evidence was largest? And if so why do you think this is, there was less cement production taking place at the plant then, or air was coming from the direction of the plant less often?
Every month has evidence of cement production, so that we have recognized that we should carry out calculations by the AIST-MM throughout the observation period. In the revised manuscript, we have added the calculations for January and April 2018. However, due to our manpower constraint, we cannot calculate longer time period than the 7 months.

Lines 157-159: "In October 2017, short-term variations in observed CO2 and d(O2/N2) were opposite in phase, and the amplitudes of some CO2 variations were larger than those of the corresponding d(O2/N2) variations. This result suggests an effect of cement production." I think these two sentences don't join together properly. CO2 & O2/N2 opposite in phase doesn't suggest cement production, CO2 increasing and O2/N2 staying the same would suggest cement production.
Lines 222-226: The sentences have been rewritten to make the meaning clearer as "In October 2017, short-term variations in observed $CO_2$ and $\delta(O_2/N_2)$ were opposite in phase, and the amplitudes (in $\mu mol\ mol^{-1}$) of some $CO_2$ variations were larger than those of the corresponding $\delta(O_2/N_2)$ variations (Fig. 5). If the short-term variations driven by terrestrial biospheric activities and the consumption of gas, liquid, and solid fuels, then the amplitudes of $CO_2$ should be smaller than those of the $\delta(O_2/N_2)$. Therefore, this result suggests an effect of cement production superimposes on fossil fuel combustion and/or terrestrial biosperic activities".

Line 168: Change "land biospheric" to "terrestrial biospheric" as that is what you have used everywhere else.

Line 230: The words "land biospheric" have been changed to "terrestrial biospheric", as suggested.

Lines 191-192: Used "however" twice in two sentences.

Lines 164-168: We have modified the sentences to use "however" only one time.

Line 208: Change "This means CO2 presumably as well" to ""This means CO2 is presumably released as well".

Line 247: The words "This means $CO_2$ presumably as well" have been changed to "This means $CO_2$ is presumably released as well", as you pointed out.

Line 213: "CO2cement".

Line 252: The word "$CO_{2cement}$" has been changed to "$y(CO_2, cement)$".

Summary: Articles usually include something about "next steps", how the research could be developed in the future.

Also say something about the limitations of the study. Although the limitations can go in the results and discussion section if you think it will fit better there

Lines 319-325: Following sentences have been added to show limitations of the study and suggest next step. "As a remaining topic, we point out the fact that detail variations in the $CO_2$ amount fraction were not reproduced by the AIST-MM enough. This is due to insufficiency of spatial resolution of the AIST-MM at least partly, to reproduce air transport from a point source such as the cement plant in the present study. Therefore, as a next step, we should use higher-resolution atmospheric transport model to improve an agreement between the observed and simulated $CO_2$ amount fractions. It is also needed to develop more accurate method to extract $y(CO_2^*)$ due only to cement production especially for the period air-sea $O_2$ flux is substantial. Such improvement will make it possible to estimate amounts of $CO_2$ capture and/or $CO_2$ leak around the observation site from an inversion analysis using the higher-resolution atmospheric transport model".

Line 284: In the acknowledgments change "observation" to "observations".

Line 359: The word "observation" has been changed to "observations", as suggested.

Lots of Figures: Figure 4 and Figure 5 are actually 5 figures each, (a-e) for each of the months. I think this is probably too many figures. Could you try combining them in some way, or choose an example month and move the others to the supplement.

We have chosen an example month and moved the others to the supplement, following your comments.

In Figure 2 and the top panels of the Figure 4's CO2 is in units of µmol mol-1. Isn't this just ppm units, that is what most people are more familiar with?
We recognize "ppm" you suggested is more familiar with our research field, however, I have used the unit following the Editor's comment.

Figure 2 Caption: Change "1-week average" to "1-week rolling average"
Figure 2 Caption: The words "1-week average" have been changed to "1-week rolling average".

Figure 2 Caption: Add something about how the CO2 & O2 y-axes are scaled to be visually comparable or the O2 y-axis is 5 times larger than the CO2 y-axis or something like that.
Figure 2 Caption: The sentence "$\delta(O_2/N_2)$ and $CO_2$ y-axes are scaled to be visually comparable" has been added, as suggested.

Figure 2: Could you add another panel for the Oxidative Ratio. I know we can see it for some of the individual months but I'm curious to see it for the whole time period.
Figure 3(b): The figure to show the Oxidative Ratio for the whole period has been added as Fig 3b.

The supplement doesn't include any of the model output or the CO measurements.
The CO amount fraction data can be found at the WDCGG. Model outputs of $CO_2$ amount fraction have been added to the supplement.

---

## Author Comment (AC2)

**Responses to Referee 2**

The objective of this paper is to examine if it is possible to detect a CO2 signal at a measurement station that is coming from a cement plant 6 km away, and to extract this specific atmospheric signal using continuous O2and CO2 measurements. It is important to have the ability to distinguish between different contributors to the atmospheric signal of CO2, which gives the opportunity to study different carbon sources and sinks separately and verify CO2 emissions. This paper studies the short term relationship between O2 and CO2 and their resulting OR to detect a signal from the cement plant. The low OR signals that originate from air with a high CO concentration show that indeed a CO2 signal of the cement plant at this measurement location can be detected. By subtracting the CO2 signals of fossil fuel combustion and the biosphere from the total atmospheric CO2 signal, based on their combined OR signals, the variations in the CO2 signal caused by only the cement plant were shown with both the measurements and a regional atmospheric transport model. These results show the ability to use the relationship between O2 and CO2 to validate CO2 fluxes from a cement plant in a transport model and to use O2 as an indicator of possible leakages of carbon capture and storage locations.

This paper shows interesting and innovative results on how O2 can be used in this context and to validate models. This work is very relevant, as studies using atmospheric O2 are scarce and therefore there is much to be learned about this tracer. This study builds on previous work by e.g. Keeling et al. (2011), van Leeuwen and Meijer (2015) and Pak et al. (2016) and gets a step closer to understanding how the ratio between O2 and CO2could be used to detect leakages from carbon capture storage locations. This is done by combining data with models, which has not often been done before with atmospheric O2. I therefore find this study of importance and would recommend it for publication, taking into account the comments below. These are mainly focussed on clarification of the results, figures and the assumptions that are made in the paper.

Thank you very much for your significant and useful comments on the paper "Measurement report: Method for evaluating $CO_2$ emissions from a cement plant using atmosphere $\delta(O_2/N_2)$ and $CO_2$ measurements and its implication for future detection of $CO_2$ capture signals" by Ishidoya et al. We have revised the manuscript, considering your comments and suggestions. Details of our revision are as follows. The line numbers denote those of the revised manuscript.

Major comments:

In line 31 the term exchange ratio (ER) is introduced as oxidative ratio (OR). However, OR is not correctly in all contexts used in the manuscript, as for example it does not apply to photosynthesis as O2 is produced there. I would recommend using ER instead. Note that there are several terms in use in the O2 community that all indicate the link between CO2 and O2 but on a different scale/process (e.g. ER, OR, alpha_B, ARQ).

The word "oxidative ratio" and "OR" have been changed to "exchange ratio" and "ER", respectively, throughout the paper, following your suggestion.

Furthermore, I would recommend to add further clarification about combining OR signals of different processes where the flux sign of O2 and CO2 are opposite. For example, in line 150 it is stated that a lower ER than 1.1 is observed and therefore shows an influence of the cement plant (which as an OR of 0). As this is probably the case, because the CO concentration is also high with these lower OR signals, I still think it is important to discuss what could happen when fluxes with different ER mix and that a ER lower than 1.1 does not directly indicate that a process is contributing with an ER lower then 1.1. When for example air from the biosphere (depleted in CO2, high in O2 and ER of 1.1) mixes with air that is mainly influenced by fossil fuel (high in CO2, depleted in O2 and ER around 1.4) you do not necessarily get an averaged ER of $(1.1 + 1.4)/2 = 1.25$ or necessarily between 1.1 and 1.4. With a large photosynthesis signal the ER could potentially even become lower than 1.1, whereas with a large fossil fuel signal, the ER would more likely be in between 1.1 and 1.4.

Lines 196-209, Fig. 3b, c: The sentences and figures have been added to discuss about combining OR signals of different processes. The sentences are as follows.

"We also plotted the ER values calculated by least-squares fitting of regression lines to the observed $\Delta y(O_2)$ and $\Delta y(CO_2)$ values during successive 24-h periods in Fig. 3b. As seen in the figure, both ER values higher and lower than 1.1 were observed throughout the observation periods. When terrestrial biosphere emits $CO_2$ to the atmosphere, i.e. respiration signal is larger than photosynthesis signal, the ER values ranging from 1.05 to 2.00 are expected from combination fluxes of terrestrial biospheric activities, gas, liquid, and solid fuels combustion. Similar ER values have been observed at other Japanese sites (e.g. Minejima et al., 2012; Goto et al., 2013; Ishidoya et al., 2020).

On the other hand, when photosynthesis signal is larger than respiration signal, ER for the combination fluxes could be variable and potentially even become lower than 1.05. However, we consider the observed low ER values are attributed to substantial $CO_2$ flux from cement production, of which ER value is 0, rather than the photosynthesis signal because the low ER values and high $\Delta y(CO)$ appeared simultaneously. These characteristics can be seen from the typical ER and $\Delta y(CO)$ in August 2018 plotted in Fig. 3c. Therefore, it is considered that the ER lower than 1.05 indicates $CO_2$ flux from cement production mixes with the surrounding air that has already been influenced by terrestrial biospheric activities or fossil fuels combustion. Similar characteristic relationships have previously been observed only in artificial $CO_2$ release experiments of which ER value is 0, such as those described by van Leeuwen and Meijer (2015) and Pak et al. (2016)."

Another point in the text where this applies is equation 4, where alpha_B+F is indeed an ER of the atmosphere without cement production (line 186), but not as the term seems to indicate an average of

the ER of the biosphere and fossil fuel.

Lines 161-163 and Lines 233-235: The $\alpha_{B+F}$ is not an average of the ER of the biosphere and fossil fuel, but monthly average ER values calculated from the simulated $O_2$ and $CO_2$ values without considering the contribution of cement production. Therefore, we have modified the sentences as follows. "For $\alpha_{B+F}$ values, we use monthly average ER values calculated from the simulated $O_2$ and $CO_2$ values without considering the contribution of cement production (black dotted line in Fig. 5, bottom, discussed below)" and "Both the observed ER values and those simulated are frequently lower than 1.1, while the ER values simulated without including cement production show lower values than 1.1 occasionally (Fig. 5 and Fig. A1a-f in Appendix A)".

In line 172 it is also not clear to me how the authors converted. From the text it seems that the atmospheric mole fractions of CO2 are converted to O2 with the ER. However, these relationship between CO2 and O2 are for the surface fluxes. Could you please specify how the ER based on the surface fluxes or process level could relate directly to the atmospheric mole fractions?

Lines 147-149: The sentences have been rewritten as follows to make the meaning clearer. "For this purpose, $O_2$ amount fractions are calculated by summing up the respective contributions of $CO_2$ amount fractions for fossil fuel combustion, terrestrial biospheric activities, and cement production multiplied by the –ER values of –1.4, –1.1, and 0. Here the 1.4 and 1.1 are typical ER for fossil fuel combustion and terrestrial biospheric activities, respectively".

Overall, I do not think something is necessarily wrong in the method, but the formulation could be more precise and a discussion about mixing different atmospheric ER signals could possibly be added. A validation of the atmospheric transport model and with that the input of the fluxes, together with a validation of the data itself is currently missing. For example, in line 234 it is stated that the complex topography can influence the model results in this area for February 2018. It is not clear why this is only the case in this month, and it would be good to see further details and validation.

We have found a mistake in the analysis of the $y(CO_2{}^*)$ in February 2018. In the revised manuscript, discrepancy between the monthly means of $y(CO_2{}^*)$ anomalies and $y(CO_2, cement)$ is not so serious, so that we have removed the sentence you pointed out.

In line 162-165 it is stated that the observed and modelled CO2 amount fractions showed weak correlation and that the general characteristics are observed but not the phase and the amplitude. This is not visible in Figure 4. Could you please elaborate more on this? Maybe by showing a graph that shows the relationship between CO2 modelled and observed?

Lines 212-221 and Fig. 4: Following sentences have been added to discuss validity of the atmospheric transport model. "Figure 4a shows monthly average of hourly $CO_2$ amount fraction is slightly

overestimated at night and underestimated in the daytime except for February, however, absolute value of the difference is less than 2 μmol mol$^{-1}$ in most case. Figure 4b is a scatter plot of the difference from 391.14 μmol mol$^{-1}$ (the minimum concentration of observed $CO_2$ in the7-months) between calculated and observed concentration for all the hourly data in the seven months. FAC2 (fraction of calculations within a factor 2 of observations) is 0.976, where model acceptance criterion of FAC2 is greater than 0.5 (Hanna and Chang, 2012), and Pearson's correlation coefficient is 0.69. The discrepancies between observed and simulated values can be attributed to the limited resolution of the model in the complex terrain, or to problems in the parameterization of transport processes, or in the $CO_2$ sources/sinks incorporated into the AIST-MM".

In line 210-215, it is stated that y(CO2*) could be used to validate this transport model. However, I miss here a discussion/validation how accurate y(CO2*) is before it could be used to validate the model. Is there a way to validate how accurate the O2 method is to extracting the cement signal from the CO2 atmospheric signal? This would help strengthen the argument that this O2 based methods works well to capture such a signal.

Lines 270-281, 338-341 and Fig. A3: We consider it is difficult to validate the $O_2$ method itself directly. Instead, we have expanded the discussion about a comparison between the observed $y(CO_2^*)$ anomalies and simulated $y(CO_2, cement)$ as follows. "We have also confirmed monthly mean $y(CO_2, cement)$ values were related to the occurrence of northwesterly winds (i.e. wind blowing from the cement plant). However, the average wind direction simulated by the AIST-MM when high $y(CO_2, cement)$ values appeared (around 300°) was slightly but systematically different from that for observed wind direction (around 270°) (Fig. A3a and A3b in Appendix A). This discrepancy is probably due to the underestimation of the altitude of Ryori ridge which locates between the cement plant and the RYO site. Such the underestimation makes it easy to transport the $CO_2$ emitted from the cement plant directly to RYO over the ridge since the cement plant is located around 300° from the RYO site. This is also consistent with the fact that the larger monthly mean $y(CO_2, cement)$ than the monthly mean $y(CO_2^*)$ anomalies are found in January and February when prevailing wind direction is northwesterly. The complex terrain around RYO such as Ryori ridge would also contributes to the discrepancy between the monthly mean $y(CO_2^*)$ anomaly and $y(CO_2, cement)$ in May and August at least partly. In May, it is considered that an effect of the oceanic $O_2$ flux on $y(CO_2^*)$ anomaly is also substantial, since we can distinguish short-term variations in $\delta(O_2/N_2)$ without simultaneous changes in $CO_2$ amount fraction (Fig. A1e)."

Something that was not clear for me, was why a baseline was subtracted from y(CO2*)? Was this done to exclude the effect of the ocean? If so, does this mean that the ocean signal was already excluded in equation 4 (to calculate y(CO2*)) by using the Δ values of CO2 and O2? If this was not the case, does this mean that the results of Δy(O2) and Δy(CO2) are still affected by the ocean and that for example

Figure 3 should be interpreted more carefully as in line 222 it is given that ocean exchange can significantly influence the observations? Could you please elaborate on this and indicate more precisely why for both y(CO2*) and Δy(CO2) a baseline is subtracted? And add further discussion on the influence of the ocean exchange on the results?

Equation 4 and Fig. 6: We have removed "Δ" from eq. 4 to avoid confusion, considering your comments. In this regard, we use Δ in Fig. 6, as the meaning shown in the caption: "Variations in $\Delta y(CO_2^*)$ calculated from the observed $CO_2$ amount fractions and $\delta(O_2/N_2)$ (black filled circles) in October 2017, and the baseline variation (blue solid line). Δ denotes deviations from their monthly mean values."

The terms Dy(O2) and Dy(CO2) and y(CO2*) are not clear, and especially the 'y' is not clearly explained and this can lead to confusion for the reader. I would recommend not using these terms and changing this throughout the manuscript, as it makes the paper more difficult to read very quickly or to interpreted the figures on their own. Also, the definition used now does not always seem consistent, as e.g. in Figure 4 the top and middle panels y-axis are both y(CO2), but these do not have the same units. Maybe the current abbreviations that indicate the different kind of CO2 signals could be changed into abbreviations that are more distinguishable. For example, the CO2,cement is more clear

We understand your suggestion, however, I have used the "$y$" following the Editor's comment. At lines 157-158, we describe the meaning of $y$ as "Here, $y$ stands for the dry amount fraction of gas, as recommended by the IUPAC Green Book (Cohen et al., 2007)". The middle panel y-axis in Fig. 5 (Fig. 4 in ACPD) has been changed to $\Delta y(CO_2)$, considering your comments.

There are quite some subplots in each figure and not every subplot is indicated with a letter or legend. This makes reading the figures confusing. Next to that, the amount of subfigures for each month makes it difficult to see all the details. For example, the statements in lines 157 and 193 are difficult to see in the figures. I also think the monthly figures do not contribute to the story. I would recommend moving part of Figure 4 and 5 in the appendix and only focus on one month to make your conclusions from them more clear.

We have chosen an example month and moved the others to the supplement and added needed legends to all the figures, following your suggestion.

Minor comments:

Title: the title of this paper could be improved. I do not think this paper is a measurement report, but rather a new method to detect cement signals. Also, the authors do not apply this method to detect carbon capture signals. It would be good to remove these points from the title and focus it in the core of the paper which is detecting cement signal.

We understand your suggestion, however, I wrote the phrase "measurement report" following the Editor's comment. Therefore, we have revised the title considering your suggestion as follows: "Measurement report: Method for evaluating $CO_2$ emissions from a cement plant using atmospheric $\delta(O_2/N_2)$ and $CO_2$ measurements and its implication for future detection of $CO_2$ capture signals".

Line 10: I would recommend using $\delta(O2/N2)$ instead of O2/N2 ratios (throughout the manuscript).
The words $O_2/N_2$ ratio have been changed to $\delta(O_2/N_2)$ throughout the paper, as suggested.

Line 14: please change 'amount fraction' to mole fraction (throughout the text).
We recognize "mole fraction" you suggested are more familiar with our research field, however, I have used the phrase following the Editor's comment.

Line 43: Friedlingstein et al. (2020) should be updated to Friedlingstein et al. (2022).
Line 44: "Friedlingstein et al. (2020)" has been updated to "Friedlingstein et al. (2022)", as suggested.

Line 43-44: The value given for the contribution of cement to the global fossil fuel CO2 emission (4%), is not correct, and is 2% for the recent decade. Also, this value is not based on atmospheric O2/N2 ratios as suggested in the text by the reference to Manning and Keeling, 2006.
Line 44: "about 4 % of… " has been changed to "about 2 % of… ", as you pointed out. The words "…and this value is included in global $CO_2$ budget analyses based on the atmospheric $O_2/N_2$ ratio (e.g. Manning and Keeling, 2006)" have been deleted to avoid confusion.

Line 52: 'Leeuwen and Meijer' should be 'van Leeuwen and Meijer'.
Line 53: "Leeuwen and Meijer" has been changed to "van Leeuwen and Meijer".

Line 70: Please specify at what height the measurements were taken and what the surface below the measurement tower consists of, and include references to previous work of the O2 measurements done here, including e.g. the precision and accuracy of the measurements etc.
Line 73 and 95, lines 92-111: The altitude of the RYO is 260 m a.s.l. (line 73). Sample air was taken at the tower heights of 20 m using a diaphragm pump (line 95). The sentences to describe the details of the $O_2$ measurements have been added (lines92-111).

Methods section: Some details were missing in the methods, but were eventually discussed in the results. For example: the methods to determine if a cement signal was seen in the data and how the cement signal was extracted from the model/data (lines 179-199 and equations 4 and 5). Please move this to the methods.

Lines 152-174: The sentences and equations you pointed out have been modified and moved to methods section.

Line 96: How was the reproducibility of 5 per meg determined? Please specify.

Lines 106-108: The sentence has been modified to describe how we determined the reproducibility as "The analytical reproducibility of the $\delta(O_2/N_2)$ and $CO_2$ amount fraction measurements by the system was determined by repeated measurements of standard gas and found to be about 5 per meg and 0.06 $\mu mol\ mol^{-1}$, respectively, for 2-minute-average values".

Please include which WMO scale was used (X2019?)?.

Line 114: The words "WMO scale" have been changed to "the WMO scale (X2007)".

Line 111: Can you include the domain in figure 1?.

Figure 1: The inner and outer domains have been included in the figure.

Line 145: Why did you choose for 1-week to subtract from the measurements? How did you determine this specific time frame?

Lines 192-194: The period is not necessarily to be 1-week, but it should be longer than short-term variations due to local effects of cement production. We have modified the sentences as follows to make the meaning clearer. "In this study, we focused on the short-term variations in $\delta(O_2/N_2)$ and the $CO_2$ and CO amount fractions (Fig. 2) to extract local effects of cement production. Therefore, we subtracted 1-week rolling average values of $\delta(O_2/N_2)$ and the $CO_2$ and CO amount fractions from the observed values to exclude their baseline variations…".

Line 145-149: It is not clear to me how the authors reached this conclusion. How many points were used to determine the OR signals that could be seen in Figure 3? Are these lines based on only 2 values? Could you please specify this?

Lines 193-201, Fig. 3a, b: The sentences have been rewritten as follows to make the meaning clearer. "Therefore, we subtracted 1-week rolling average values of $\delta(O_2/N_2)$ and the $CO_2$ and CO amount fractions from the observed values to exclude their baseline variations, and examined the relationships among the residuals ($\Delta y(O_2)$, $\Delta y(CO_2)$, and $\Delta y(CO)$; Fig. 3a). Here, $\Delta y(O_2)$ is the equivalent value in $\mu mol\ mol^{-1}$ converted from $\delta(O_2/N_2)$. We also plotted the ER values calculated by least-squares fitting of regression lines to the observed $\Delta y(O_2)$ and $\Delta y(CO_2)$ values during successive 24-h periods in Fig. 3b. As seen in the figure, both ER values higher and lower than 1.1 were observed throughout the observation periods. When terrestrial biosphere emits $CO_2$ to the atmosphere, i.e. respiration signal is larger than photosynthesis signal, the ER values ranging from 1.05 to 2.00 are expected from

combination fluxes of terrestrial biospheric activities, gas, liquid, and solid fuels combustion. Similar ER values have been observed at other Japanese sites (e.g. Minejima et al., 2012; Goto et al., 2013; Ishidoya et al., 2020)."

Line 163: Are these the monthly average correlations?

We have removed the related sentence. Instead, the sentences have been added to discuss validity of the atmospheric transport model (Lines 212-221 and Fig. 4).

Line 190-192: How valid is your assumption that ocean fluxes are not influencing the results?

We cannot validate it completely. However, as we described in lines 164-174, $y(CO_2^*)$ anomaly obtained by subtracting the baseline variation is considered to indicate $CO_2$ changes due mainly to the contribution of the cement production since temporal variations in $\delta(O_2/N_2)$ due to the contribution of oceanic signal are generally slower than that of the cement production.

Line 208: Does this statement mean that you miss some of the CO2 signal of the cement plant in Figure 5? Please specify.

Lines 248-249: The sentence has been rewritten as follows to make the meaning of "overall ER" clearer. "This means $CO_2$ is presumably released as well, so that the overall ER for the $CO_2$ emitted from cement plant (cement production + fossil fuel combustion) would not be 0."

Line 222: Here, it is mentioned that the ocean fluxes can significantly influence the observed signals. See the major point above, and my comment at line 190-192, and please address this point in the discussion of the paper.

As we mentioned above, we cannot validate it completely whether the oceanic flux effect is excluded enough or not. This is a limitation of our method, so that we have add sentences "Therefore, as a next step, we should use higher-resolution atmospheric transport model to improve an agreement between the observed and simulated $CO_2$ amount fractions. It is also needed to develop more accurate method to extract $y(CO_2^*)$ due only to cement production especially for the period air-sea $O_2$ flux is substantial" (lines 321-324). We consider the oceanic flux effect is substantial in May, 2018 since we can distinguish short-term variations in $\delta(O_2/N_2)$ without simultaneous changes in $CO_2$ amount fraction (Fig. A1e) (lines 279-281).

Line 234: Why is the complicated topography only a problem in February 2018 and not in other months? And can this fully explain the difference between simulated and observed signals, also for other months? This issue needs more explanation.

As we replied to your major comments, we have found a mistake in the analysis of the $y(CO_2^*)$ in

February 2018. In the revised manuscript, discrepancy between the monthly means of $y(CO_2^*)$ anomalies and $y(CO_2, \text{cement})$ is not so serious, so that we have removed the sentence you pointed out.

Lines 241-247: The link between the method presented here to detect the cement plant emissions and detection of leakages from carbon capture sites is made several times throughout the paper. During this study it is made clear that with the help of CO we could see if the air came from fossil fuel sources or the cement plant. However, there would be no source of CO when the method is applied to detect carbon capture leakages. The method would work for carbon capture from a flue gas (line 60). I think it is good to make a distinction of when CO needs to be used, as it is quite an important component of this research.

Lines 288-290: The sentences "It should be also noted that we did not use CO amount fraction for the calculation of $y(CO_2^*)$. This is an important advantage to apply $y(CO_2^*)$ to detect $CO_2$ capture and/or $CO_2$ leak which do not emit CO." have been added to indicate the method works without help of CO.

Line 217: I wonder if there is a way to go from the CO2 anomalies caused by the cement plant (figure 5) to the emissions of the cement plant. As this could be a crucial step to use this approach for emission verification. Could you discuss this?

Could you please separate the results and discussion sections, including several subsections, and rewrite the summary section to a conclusion section?

Lines 319-325: Following sentences have been added to discuss future tasks needed to estimate the emission of the cement plant from the observed and simulated $CO_2$ anomalies.

"As a remaining topic, we point out the fact that detail variations in the $CO_2$ amount fraction were not reproduced by the AIST-MM enough. This is due to insufficiency of spatial resolution of the AIST-MM at least partly, to reproduce air transport from a point source such as the cement plant in the present study. Therefore, as a next step, we should use higher-resolution atmospheric transport model to improve an agreement between the observed and simulated $CO_2$ amount fractions. It is also needed to develop more accurate method to extract $y(CO_2^*)$ due only to cement production especially for the period air-sea $O_2$ flux is substantial. Such improvement will make it possible to estimate amounts of $CO_2$ capture and/or $CO_2$ leak around the observation site from an inversion analysis using the higher-resolution atmospheric transport model."

We have changed the "summary" section to a "conclusions" section, as suggested. We leave the "results and discussion" section as it is, since we think the reader can follow it without separating into subsections. Instead, we have moved some sentences to the method section from the results and discussion section, as you suggested above, and separated the method section into subsections.

---

## Author Response (AR2)

**Responses to Referee 1**

I think this is a good article and I'm happy with the authors response to my comments. As I was checking though this time I spotted a few typos that should be corrected, and I also think the final paragraph could benefit from being reworded. Other than that I think it can be published.

Thank you very much for your significant and useful comments on the paper "Measurement report: Method for evaluating $CO_2$ emissions from a cement plant using atmosphere $\delta(O_2/N_2)$ and $CO_2$ measurements and its implication for future detection of $CO_2$ capture signals" by Ishidoya et al. We have revised the manuscript, considering your comments and suggestions. Details of our revision are as follows. The line numbers denote those of the revised manuscript.

Line 10: I think "Continous observations of the atmospheric" should be "Continous observations of atmospheric"

Line 10: The words "Continuous observations of the atmospheric" have been changed to "Continuous observations of atmospheric", as suggested.

Line 18: I think "$O_2$ and $CO_2$ amount fractions changes" should be "$O_2$ and $CO_2$ amount fraction changes"

Line 18: The words "$O_2$ and $CO_2$ amount fractions changes" have been changed to " $O_2$ and $CO_2$ amount fraction changes ", as suggested.

Line 21: I'm pretty sure "OR" should be "ER"

Line 21: The "OR" has been changed to "ER".

Line 216: This line is missing a space "the7-months"

Line 229: The words "the7-months " have been corrected to "the 7-months ". Thank you for pointing it out.

Line 217: Should probably change "seven months" to "7 months" as the number is used everywhere else

Line 230: The words "seven months" have been changed to "7-months ".

Line 223/224: I think "If the short-term variations driven" should be "If the short-term variations were driven"

Line 236-237: The words "If the short-term variations driven" have been corrected to "If the short-term variations were driven".

Line 226: "biosperic" should be "biospheric"

Line 239: The typo "biosperic " has been corrected to "biospheric ".

Line 239: I think "(see 2.3 in details)" should be "(see details in section 2.3)"

Line 252: The words "see 2.3 in details" have been changed to "see details in section 2.3".

Line 259: "Februaty" should be "February"

Line 272: The typo "Februaty" has been corrected to "February".

Line 263: I'm pretty sure "OR" should be "ER"

Line 276: The "OR" has been changed to "ER".

Line 286: I think "which locates between" should be "which is located between"

Line 287: The words "which locates between" have been changed to "which is located between".

Line 290: I think "also contributes to" should be "also contribute to"

Line 291: The words "also contributes to" have been changed to "also contribute to".

Last paragraph: while I'm glad you've added in a limitations/next steps paragraph, I actually think it comes off as a bit negative, and that it could be improved with softer language, for example instead of saying "we need to do this" or "we should do this" say "this could be done". I rewrote this part to show what I mean.

Some of the more detailed variations in the $CO_2$ amount fractions were not reproduced by the AIST-MM. This is at least partially due to the spatial resolution of the AIST-MM which limited its ability to reproduce air transport from a point source, such as the cement plant in the present study. In the future this work could be expanded on by using a higher resolution atmospheric transport model to improve the agreement between the observed and simulated $CO_2$ amount fractions. An additional step could be developing a more accurate method for extracting $y(CO_2*)$ due only to cement production, especially for the period when air-sea $O_2$ flux is substantial. This would improve the estimation of the amount of $CO_2$ capture and/or $CO_2$ leak around the observation site from an inversion analysis using the higher-resolution atmospheric transport model.

Lines 333-340: The sentences have been rewritten following your suggestion. Thank you for editing the sentences.

Appendix A title: I think "to evaluate an effect" should be "to evaluate the effect"

Line 358: The words "to evaluate an effect" have been changed to "to evaluate the effect".

Figure 4b: The plot has "all" in the top right hand corner, which I don't think is necessary
Figure 4b: The word "all" has been removed.

Figure 4 caption: "μmol mol－1" isn't in bold and I think "both of data groups" should be "both of the data groups"
Figure 4 caption: Both words have been rewritten, as suggested.

Other point
Lines 30-31: We have added Liu et al. (2023) as a reference since they applied atmospheric $O_2$ measurements to evaluate urban $CO_2$ cycle recently.

**Responses to Referee 2**

I would like to thank the authors for taking my comments into consideration and adjusting the paper accordingly. The paper has improved significantly. The validation of the model in Figure 4 strengthened the study, by showing that the transport model performs well. The addition of some examples in the main text and adding the equations to the method sections improved the readability of the paper. The analysis of mixing different signals with specific ER signals has also improved. However, on this last point still some improvement could be made. Therefore I would like to ask the authors to take the following points into consideration.

Thank you very much for your significant and useful comments on the paper "Measurement report: Method for evaluating $CO_2$ emissions from a cement plant using atmosphere $\delta(O_2/N_2)$ and $CO_2$ measurements and its implication for future detection of $CO_2$ capture signals" by Ishidoya et al. We have revised the manuscript, considering your comments and suggestions. Details of our revision are as follows. The line numbers denote those of the revised manuscript.

There is still some discrepancy on how the term Exchange Ratio (changed from Oxidative Ratio of the previous paper) is used. Now ER is used everywhere, however fossil fuel ER values are closely linked to OR signals. I would recommend to carefully evaluate when each term should be used. The ER should be used when describing measurements in the atmosphere and what is exchanged between the surface and the atmosphere, the OR is more related to the processes/oxidation of the fossil fuel burning itself.

Lines 37-41: Following sentences have been added considering your comments.

"It is noted that the words "Oxidative Ratio (OR)" has also been widely used in by the same definition as ER. Strictly speaking, there is a distinction in the terminology between ER and OR; the ER indicates the exchange between the atmosphere and organisms or ecosystems while the OR indicates the stoichiometry of specific materials (Faassen et al., 2023). We use the ER throughout the present study conveniently, but the distinction should be kept in mind."

Line 211 of track changes document: The budget of the amount fractions of O2 and CO2 at RYO are determined by adding up the terrestrial biosphere activity, fossil fuel combustion and cement production. By focussing on short time scales the authors exclude the ocean and the three previously mentioned processes should cover the complete budgets of O2 and CO2. However, because the authors look at hourly time scale I wonder if entrainment of air from the free troposphere into the atmospheric boundary layer makes the budget calculation of O2 more complicated. Entrainment of O2 is not linked to an ER signal of one of the three previously mentioned budget processes, but the ER is rather a result of the difference between the boundary layer (mainly biosphere) and the free troposphere (combination

of fossil fuel and biosphere). This would result in a decoupling of the ER signals between the boundary layer and the free troposphere and can create a difference in behaviour of O2 compared to CO2 of the entrainment flux that cannot be linked to one specific ER signal. I am not sure if this could be easily checked. Would it be possible for the authors to add the validation of O2 from the transport model with the observations and see if there is a mismatch during the morning transition? Or would it be possible to transport O2 by linking the ER signals directly to the surface fluxes of CO2? This should make sure that the budget of O2 at RYO is calculated correctly.

Lines 154-164 and 342-356, and Fig. A1a-d: Following sentences have been added considering your comments. Figures A1a-d have also been added.

"In this regard, it should be noted that Faassen et al. (2023) carried out continuous observations of $\delta(O_2/N_2)$ and the $CO_2$ amount fraction at a forest site in Finland, and they found higher ER (referred to as "$ER_{atmos}$" in their study) than 2.0 during the morning transition for the average diurnal cycle in summer. Such high ER cannot be obtained from summing up the contributions of fossil fuel combustion and terrestrial biospheric activities at the surface, so that they suggested the ER signal not only represented the diurnal cycle of the forest exchange but also includes other factors, including entrainment of air masses in the atmospheric boundary layer before midday, with different thermodynamic and atmospheric composition characteristics. Considering their results, we examined average diurnal cycles of $\delta(O_2/N_2)$ and the $CO_2$ amount fraction at RYO in October 2017 and August 2018 (Fig. A1a-d in Appendix A). We found the ER values are close to 1 throughout the day both for the observed and simulated diurnal cycles. Therefore, we consider the entrainment of air masses do not change the ER at RYO substantially, and the atmospheric transport processes in the AIST-MM is appropriate to compare the observational results in the present study."

"Appendix A: Additional figures to evaluate the effect of entrainment of air mass on the observed ER
As we described in 2.2, Faassen et al. (2023) found higher ER ("$ER_{atmos}$" in their study) than 2.0 at a forest site in Finland during the morning transition for the average diurnal cycles of $\delta(O_2/N_2)$ and the $CO_2$ amount fraction in summer. On the other hand, Ishidoya et al. (2013) reported ER values ("$ER_{atm}$" in their study) close to 1 at a Japanese forest site in summer, for the average diurnal cycles throughout the day. Considering the discrepancy between Faassen et al. (2023) and Ishidoya et al. (2013), we derive the average diurnal cycle of $\delta(O_2/N_2)$ and the $CO_2$ amount fraction at RYO. For this purpose, deviations of $\delta(O_2/N_2)$ and the $CO_2$ amount fraction from their 24-hr mean values were calculated, and the $\Delta\delta(O_2/N_2)$ were converted to $\Delta y(O_2)$ by multiplying $X(O_2)$ (=0.2094). Figures A1a-b show the average diurnal cycles of $\Delta y(O_2)$ and $\Delta y(CO_2)$ in October 2017, and their relationship. Those for August 2018 are also shown in Figs. A1c-d. As seen from the figures, the observed $\Delta y(O_2)$ took maxima in the daytime, and the ER values for the average diurnal cycles at RYO were close to 1 throughout the day. The corresponding diurnal $\Delta y(O_2)$ and $\Delta y(CO_2)$ cycles and their relationships obtained from the simulated results by AIST-MM were also shown in Figs. A1a-d. Similar to the

observations, it was found that the simulated $\Delta y(O_2)$ took maxima in the daytime and the ER were close to 1 throughout the day. These facts indicate the observed ER at RYO can be reproduced by the AIST-MM generally, including the period during the morning transition. Therefore, an entrainment of air mass to yield high ER during the morning suggested by Faassen et al. (2023) may be a characteristic phenomenon at their observational site."

Line 274 of track changes document: I would recommend to re-evaluate this sentence; 'Therefore, it is considered that the ER lower than 1.05 indicates CO2 flux from cement production mixes with the surrounding air that has already been influenced by terrestrial biospheric activities or fossil fuels combustion.' The number 1.05 seems like a hard limit for me, because combining different processes of the biosphere and fossil fuels could also produce ER values lower than 1.05 and it seems that Figure 3 and 5 also show this. I appreciate that the authors added the discussion that combing the processes of photosynthesis and respiration could create an ER that is lower than 1.05. However they argue that this is not the case because ER signals lower than 1.05 always show a correlation with high CO concentrations. Based on Figure 3 and 5, I would argue that ER signals lower than 1.05 could also indicate mixing of fossil fuels with the biosphere or even changes in the ER signal of the biosphere itself because of still high correlation coefficients (Figure 5) or no peak of CO (Figure 3) during some events where the ER drops below 1.05. Adding a fossil fuel signal to a dominant biosphere signal produced by the surface could also create low ER signals (and not an ER between 1.1 and 1.4) because of mixing processes with an opposite flux for O2 and CO2 (same explanation as given in line 270). The high CO concentration would then indicate a fossil fuel signal rather than a cement signal. I would recommend to add the criteria that the correlation coefficient between O2 and CO2 should be lower than a specific value in combination with an ER signal lower than 1.05 as an indication for a cement production signal.

Lines 215-221 and Fig. 3c: Following sentences have been added considering your comments. $\Delta y(CO_2)$ values have also been added to Fig. 3c.

"On the other hand, when photosynthesis signal is larger than respiration signal, ER for the combination fluxes could be variable and potentially even become lower than 1.05. Therefore, we consider the observed low ER values with high $\Delta y(CO)$ and $\Delta y(CO_2)$ are attributed to substantial $CO_2$ flux from cement production, of which ER value is 0, rather than the photosynthesis signal. These characteristics can be seen from the typical ER, $\Delta y(CO)$ and $\Delta y(CO_2)$ in August 2018 plotted in Fig. 3c. Therefore, it is considered that the air mass having ER lower than 1.05 and $\Delta y(CO)$ and $\Delta y(CO_2)$ higher than 0 simultaneously indicates $CO_2$ flux from cement production mixes with the surrounding air that has already been influenced by terrestrial biospheric activities or fossil fuels combustion"

Figure 3c and Figure 5 bottom: The authors explain in the text that the ER signals in these figures are based on regression lines of successive 24-hour periods. It is not entirely clear to me what this means and what kind of implications this would have for the ER signals in the figures. Does successive mean a linear regression of the previous 24 hours of each point? If that is the case, then figure 3c and the bottom figure of 5 would give a false idea of an hourly ER signal because this figure looks at a 24 hour signal. Could this be clarified?.

In the manuscript, I did not write these figures give an hourly ER. The ER signals were 24 hour signal, obtained as slopes of the regression lines to the observed data during successive 24-h periods (before and after 12-h of each point). Therefore, the words "24-h periods (before and after 12-h of each point)" have been added to the caption of Fig. 3c.

Figure 4a: I highly appreciate the addition of this figure that validates the transport model. A minor recommendation would be to put 2017 on top of 2018 in the legend.

Figure 4a: The legends have been modified, as suggested.

Figure 6 and 7: There seems to be a circularity in the validation of the modelled CO2 cement amount fractions with the observed cement CO2 amount fractions. For the observed cement CO2 amount fractions the ER signals of only the biosphere and the fossil fuel signal is used (ER(bio+ff)). However, to acquire this signal a model is needed and therefore the authors attain this ER value from the model that they try to validate. I understand that this was the only way to get the ER(bio+ff) signal, however I would recommend to add a discussion about this as this circularity could falsely improve the validation results. Especially because the modelled total ER signal does not match well with the observed total ER signal (Figure 5 bottom). Could the authors add a discussion on this?.

Figure 7 and line 615: Based on figure 7 the statement is made in line 615 that either an ER(bio+ff) of 1.1 or 1.4 could be used to estimate the observed contribution of cement. I would argue that is only on the monthly timescale is analysed in the figure, but it is still debatable on shorter time scales. The previous analysis of the paper showed that the cement signals are visible on hourly or diurnal time scales. The paper also showed that on hourly and diurnal time scales both the total CO2 amount fraction and the total ER can change drastically. When moving from a hourly/diurnal to monthly time scale, the CO2 and ER signal could smoothen and therefore the variation in the ER signal does not matter that much. On these shorter time scales it could therefore still be important to have a variable ER signal. I would therefore recommend specifying clearly for which time scales the statements apply here.

We consider following revisions will answer above 2 comments simultaneously.

Lines 300-303 and 374-378, and Figs. B4a-b: Following sentences and figures have been added to clarify that we do not need to use any simulated value to calculate $y(CO_2^*)$ for the model validation,

and this is also applicable on shorter time scales.

"This is also applicable on shorter time scales (Figures B4a and B4b in Appendix B). Therefore, we can derive the observed $y(CO_2{}^*)$ at RYO is without using any simulated value by an atmospheric transport model, and the observed $y(CO_2{}^*)$ can be used to validate hourly to annual average $CO_2$ fluxes from cement production simulated by a fine-scale atmospheric transport model."

"Figures B4a and B4b show the bottom panels of Fig. 6 and A2a, respectively, but for adding the $\Delta y(CO_2{}^*)$ calculated by using the $\alpha_{B+F}$ values of 1.4 and 1.1. As seen from the figures, several hours to day-to-day variations in the $\Delta y(CO_2{}^*)$ did not change substantially depending on the $\alpha_{B+F}$ value used to calculate $y(CO_2{}^*)$. Therefore, the contribution of cement production to the atmospheric $CO_2$ amount fraction at RYO can be estimated from the observed $y(CO_2{}^*)$ by assuming an $\alpha_{B+F}$ value of 1.1 or 1.4, not only for monthly time scale but for shorter (hourly to day-to-day) time scale."

Other point

Lines 30-31: We have added Liu et al. (2023) as a reference since they applied atmospheric $O_2$ measurements to evaluate urban $CO_2$ cycle recently.

---

## Author Response (AR3)

**Responses to the Editor**

I am satisfied that the referees' comments have been addressed in this revised manuscript and am happy to accept it for publication subject to the technical corrections listed below.

Thank you very much for your significant and useful comments on the paper "Measurement report: Method for evaluating $CO_2$ emissions from a cement plant using atmosphere $\delta(O_2/N_2)$ and $CO_2$ measurements and its implication for future detection of $CO_2$ capture signals" by Ishidoya et al. We have revised the manuscript, considering your comments and suggestions. Details of our revision are as follows. The line numbers denote those of the revised manuscript.

Referee #2 made an important point about the distinction between exchange ratio (ER) and oxidative ratio (OR). ER shouldn't be used "conveniently" when OR is the appropriate term (e.g., l. 32, 33, 46). Please amend lines 37 to 41 to

"It is noted that the term oxidative ratio (OR) has also been used to designate exchange ratios (ER). Strictly speaking, there is a distinction between ER and OR; the ER refers to the exchange between the atmosphere and organisms or ecosystems while the OR indicates the stoichiometry of specific materials (Faassen et al., 2023)."

and use the appropriate term throughout the manuscript, as also requested by reviewer #2.

Lines 31-34: The sentence has been rewritten considering your comments.

"This approach uses $-O_2$:$CO_2$ exchange ratios (ER) or oxidative ratios (OR) $(-\Delta y(O_2)\Delta y(CO_2)^{-1})$ for terrestrial biospheric activities and fossil fuel combustion. Strictly speaking, there is a distinction between ER and OR; the ER refers to the exchange between the atmosphere and organisms or ecosystems while the OR indicates the stoichiometry of specific materials (Faassen et al., 2023)"

Also, we have changed some ER to OR to use the appropriate term throughout the manuscript. Please find them in the manuscript with track changes.

l. 229 & 230: Please delete the hyphen from "7 months". A hyphen only appears when used as an compound adjective (without the 's' after month), e.g., 7-month average.

Lines 227-228: The word "7-months" has been corrected to "7 months", as suggested.

l. 348: Please change "24-hr" to "24-h", since the internationally accepted abbreviation for "hour" is h.

Line 346: The word "24-hr" has been corrected to "24-h", as suggested.

Other points

Line 112: The words "data points shown in Fig. 2 is 9221" have been changed to "data points shown

in Fig. 2 is 9220", since we have removed 1 error data from the original dataset (the original dataset is "Supplement_data.csv" posted in the ACPD and the revised dataset is "Supplement_data_r.csv" submitted previously).

Line 301: The words "should be also noted" has been changed to "should also be noted".

Reference: We have noticed Ishidoya et al. (2013) is cited in the text, so that it has been added to the reference.

I expect I will get the URL and DOI for the deposited data in the WDCGG within a week.